

# A simple calculation algorithm to separate high-resolution CH[4] flux measurements into ebullition and diffusion-derived components

Mathias Hoffmann[1,*], Maximilian Schulz-Hanke[2], Juana Garcia Alba[1], Nicole Jurisch[2], Ulrike Hagemann[2], Torsten Sachs[3], Michael Sommer[1,4], Jürgen Augustin[2]

[1]Institute of Soil Landscape Research, Leibniz Centre for Agricultural Landscape Research (ZALF) e.V., Eberswalder Str. 84, 15374 Müncheberg, Germany

[2]Institute for Landscape Biogeochemistry, Leibniz Centre for Agricultural Landscape Research (ZALF) e.V., Eberswalder Str. 84, 15374 Müncheberg, Germany

[3]GFZ German Research Centre for Geosciences, Telegrafenberg, 14473 Potsdam, Germany

[4]University of Potsdam, Institute of Earth and Environmental Sciences, Karl-Liebknecht-Str. 24-25, 14476 Potsdam, Germany

*Corresponding author:

Mathias Hoffmann (Mathias.Hoffmann@zalf.de)

[1]Institute of Soil Landscape Research, Leibniz Centre for Agricultural Landscape Research (ZALF) e.V.

Eberswalder Str. 84, 15374 Müncheberg, Germany

E-mail: Mathias.hoffmann@zalf.de

Tel.: +49(0)33432 82 327

Fax: +49(0)33432 82 280



**Abstract**

Processes driving the production, transformation and transport of methane ($CH_4$) in wetland ecosystems are highly complex. We present a simple calculation algorithm to separate open-water $CH_4$ fluxes measured with automatic chambers into diffusion- and ebullition-derived components. This helps to reveal underlying dynamics, to identify potential environmental drivers, and thus, calculate reliable $CH_4$ emission estimates. The flux separation is based on

identification of ebullition-related sudden concentration changes during single measurements. Therefore, a variable ebullition filter is applied, using the lower and upper quartile and the interquartile range (IQR). Automation of data processing is achieved by using an established R-script, adjusted for the purpose of $CH_4$ flux calculation. The algorithm was tested using flux measurement data (July to September 2013) from a former fen grassland site, converted into a

shallow lake as a result of rewetting. Ebullition and diffusion contributed equally (46 % and 55 %) to total $CH_4$ emissions, which is comparable to ratios given in literature. Moreover, the separation algorithm revealed a concealed shift in the diurnal trend of diffusive fluxes throughout the measurement period. The water temperature gradient was identified as one of the major drivers of diffusive $CH_4$ emissions, whereas no significant driver was found in case

of erratic $CH_4$ ebullition events.

**Keywords**

Wetland ecosystems, ebullition, diffusion, automatic chamber system, diurnal variability, R-script






## 1. Introduction

Wetlands and freshwaters are among the main sources for methane ($CH_4$) emissions (Dengel et al. 2013; Bastviken et al. 2011; IPCC 2013). In open-water systems, $CH_4$ is released via three main pathways: i) diffusion (including "storage flux", in terms of rapid diffusive release from methane stored in the water column), ii) ebullition and iii) plant-mediated transport (e.g., Goodrich et al. 2011; Bastviken et al. 2004; Van der Nat and Middelburg 2000; Whiting and Chanton 1996). The magnitude of $CH_4$ released via the different pathways is subject to variable environmental drivers and conditions such as water level, atmospheric pressure, temperature gradients, wind velocity, and the presence of macrophytes (Lai et al. 2012; Tokida et al. 2007; Chanton and Whiting 1995). As particularly ebullition varies in time and space (Maeck et al. 2013; Walter et al. 2006), total $CH_4$ emissions feature an extremely high spatial and temporal variability at all scales (Koch et al. 2014; Repo et al. 2007; Bastviken et al. 2004). Hence, attempts to model $CH_4$ emissions based on individual environmental drivers are highly complex. The separation of measured $CH_4$ emissions into the individual pathway-associated components is therefore crucial if aiming to identify relevant environmental drivers of $CH_4$ emissions (Bastviken et al. 2011; Bastviken et al. 2004). In consequence, the understanding of the complex processes determining the temporal and spatial patterns of $CH_4$ emissions is a prerequisite for upscaling field-measured $CH_4$ emissions to the landscape or regional scale, and thus for adequately quantifying the contribution of wetland $CH_4$ emissions to global greenhouse gas (GHG) budgets (Walter et al. 2015; Koebsch et al. 2015; Lai et al. 2012; Limpens et al. 2008).

However, field studies measuring $CH_4$ release above shallow aquatic environments or flooded peatlands generally measure total $CH_4$ emissions as a mixed signal of individual $CH_4$ emission components, released via all possible pathways (i.e. diffusion, ebullition and plant-mediated transport). Studies separately measuring temporal and spatial patterns of $CH_4$ emissions resulting only from either ebullition or diffusion are rare. Measurements of $CH_4$ ebullition can be performed using manual or automatic gas traps, as well as optical and hydro-acoustic methods (Wik et al. 2013; Maeck et al. 2013; Wik et al. 2011; Walter et al. 2008; Ostrovsky et al. 2008; Huttunen et al. 2001; Chanton and Whiting 1995), often requiring considerable instrumentation within the studied system. Diffusive $CH_4$ fluxes are commonly either derived indirectly as the difference between total $CH_4$ emissions and measured ebullition, or directly obtained based on the use of bubble shields or gradient measurements of $CH_4$ concentration differences (DelSontro et al. 2011; Bastviken et al. 2010; Bastviken et al. 2004). A graphical method to separate diffusion, steady ebullition and episodic ebullition



fluxes from the total $CH_4$ flux was presented by Yu et al. (2014), using a flow-through chamber system. However, performed at the laboratory scale for a peat monolith, measurement results as well as the applied method were lacking direct field applicability. A first simple mathematical approach for field measurements to separate ebullition from the sum of diffusion and plant-mediated transport was introduced by Miller and Oremland (1988),

who used low-resolution static chamber measurements. Goodrich et al. (2011) specified the approach using piecewise linear fits for single ebullition events. However, static thresholds determining ebullition events, as well as low-resolution measurements, limited the approach to estimates of medium and major ebullition events and prevented a clear flux separation. Therefore, $CH_4$ flux separation approaches based on manual chamber measurements with

rather low temporal resolution fail to capture the rapidly changing absolute and relative contributions of the pathway-associated flux components both in time and space (Maeck et al. 2013; Walter et al. 2006).

Hence, there is a need for a non-intrusive method for separating pathway-associated $CH_4$ flux components both in time and space. Improvements in measurement techniques, particularly

by using high-resolution gas analyzers (e.g., within Eddy Covariance (EC) measurements), allow for high temporal resolution records of $CH_4$ emissions (Schrier-Uijl et al. 2011; Wille et al. 2008). Recently, a growing number of experimental GHG studies employ automatic chambers (AC) (Koskinen et al. 2014; Lai et al. 2014; Ramos et al. 2006), which can provide flux data with an enhanced temporal resolution and capture short-term temporal (e.g., diurnal)

dynamics. In addition, AC measurements can also represent small-scale spatial variability, and thus identify potential hot spots of $CH_4$ emissions (Koskinen et al. 2014; Lai et al. 2014). AC systems therefore combine the advantages of chamber measurements and micrometeorological methods with respect to the quantification of spatial as well as temporal dynamics of $CH_4$ emissions (Savage et al. 2014; Lai et al. 2012).

Combined with a high-resolution gas analyzer (e.g., cavity ring-down spectroscopy), AC measurements provide opportunities for i) detecting even minor ebullition events, and ii) developing a statistically based flux separation approach. This study presents a new calculation algorithm for separating open-water $CH_4$ fluxes into its ebullition- and diffusion-derived components based on ebullition-related sudden concentration changes during

chamber closure. A variable ebullition filter is applied using the lower and upper quartile and the interquartile range (IQR) of measured concentration changes. Data processing is based on the R-script developed by Hoffmann et al. (2015), modified for the purpose of $CH_4$ flux





calculation and separation, thus including the advantages of automated and standardized flux estimation. We hypothesize that the presented flux calculation and separation algorithm can

reveal concealed spatial and temporal dynamics in ebullition- and diffusion-associated $CH_4$ fluxes, thus facilitating the identification of relevant environmental drivers.

## 2. Material and Methods

### 2.1 Exemplary field data

#### 140 2.1.1 Study site

Ecosystem $CH_4$ exchange was measured at a flooded former fen grassland site, located within the Peene river valley in Mecklenburg-Western Pomerania, northeast Germany (53°52´N, 12°52´E). The long-term annual precipitation is 570 mm. The mean annual air temperature is 8.7°C (DWD, Anklam). The study site was particularly influenced by a complex melioration

and drainage program between 1960 and 1990, characterized by intensive agriculture. As a consequence, the peat layer was degraded and the soil surface was lowered by subsidence. Being included in the Mecklenburg-Western Pomerania Mire Restoration Program, the study site was rewetted at the beginning of 2005. As a result, the water level was permanently above the soil surface, thus transforming the site into a shallow lake. Exceptionally high $CH_4$

emissions at the measurement site are reported by Franz et al. (2016), who measured $CO_2$ and $CH_4$ emissions using an eddy covariance system and Hahn-Schöffl et al. (2011), who investigated sediments formed during inundation. Prior to rewetting, the vegetation was dominated by reed canary grass (*Phalaris arundinacea*), which disappeared after rewetting due to permanent inundation. At present, the water surface is partially covered with duckweed

(Lemnoideae), while broadleaf cattail (*Typha latifolia*) and reed mannagrass (*Glyceria maxima*) are present next to the shoreline. However, below chambers, no emergent macrophytes were present throughout the study period.

#### 2.1.2 Automatic chamber system

In April 2013, the measurement site was equipped with an AC system and a nearby climate station (Fig. 1). The AC system consists of four transparent chambers, installed as transect from the shoreline into the lake. Chambers are made of Lexan Polycarbonate with a thickness of 2 mm and reinforced with an aluminium frame. Each chamber (volume of 1.5 m³; base area 1 m²) is mounted in a steel profile, secured by wires, and lifted/lowered by an electronically





controlled cable winch located at the top of the steel profile. All chambers are equipped with a water sensor (capacitive limit switch KB 5004, efector150) at the bottom, which allows steady immersion (5 cm) of the chambers into the variable water surface. Hence, airtight sealing as well as constant chamber volume are ensured during the study period. All chambers are connected by two tubes and a multiplexer to a single Los Gatos Fast Greenhouse Gas

Analyser (911-0010, Los Gatos), measuring the air concentration of carbon dioxide ($CO_2$), methane ($CH_4$), and water vapour ($H_2O$). To ensure consistent air pressure and mixture during measurements, chambers are ventilated by a fan and sampled air is transferred back into the chamber headspace. However, due to the large chamber volume, complete mixture of the chamber headspace took up to 30 seconds. In consequence, most peaks due to ebullition

events showed overcompensation (Fig. 3). Concentration measurements are performed in sequence, sampling each chamber for 10 minutes with a 15-second frequency once per hour. A wooden boardwalk north of the measurement site allows for maintenance access, while avoiding disturbances of the water body and peat surface.

### 2.1.3 Ancillary field measurements

Temperatures were recorded in different water (5 cm above sediment surface) and sediment depths (2 cm, 5 cm, and 10 cm below the sediment-water interface), using thermocouples (T107, Campbell Scientific). Additionally, air temperature at 20 cm and 200 cm height, as well as wind speed, wind direction, precipitation, relative humidity, and air pressure were

measured by a nearby climate station (WXT52C, Vaisala). Water table depth was measured by a pressure probe (PDCR1830, Campbell Scientific). All parameters were continuously recorded at 30-minute intervals and stored by a data logger (CR 1000, Campbell Scientific) connected to a GPRS radio modem.

### 2.2 Flux calculation and separation algorithm

$CH_4$ flux calculation was performed using a standardized R-script presented in detail by Hoffmann et al. (2015). Measured fluxes were determined using Eq. (1), where M is the molar mass of $CH_4$, $\delta v$ is the linear concentration change over time (t), A and V denote the basal area and chamber volume, respectively, and T and P represent the inside air temperature

and air pressure. R is a constant (8.3143 $m^3$ $Pa$ $K^{-1}$ $mol^{-1}$).

(1)

$$r_{CH_4} (\mu mol\, C\, m^{-2}\, s^{-1}) = \frac{M * P * V * \delta v}{R * T * t * A}$$



To estimate the relative contribution of diffusion and ebullition to total $CH_4$ emissions, flux calculation was performed twice, adjusting selected user-defined parameter setups of the used R-script (Hoffmann et al. 2015) (Fig. 3). First, the diffusive component of the flux rate ($CH_{4_{diffusion}}$) was calculated based on a variable moving window (MW) with a minimum size of 5 consecutive data points. Abrupt concentration changes within the MW were identified by means of a rigid outliner test, discarding fluxes with an inherent concentration change larger or smaller than the upper and lower quartile $\pm$ 0.25 times the interquartile range (IQR). Tests of variance homogeneity and normal distribution were applied with $\alpha=0.1$. Second, the total $CH_4$ flux ($CH_{4_{total}}$) for each measurement was calculated as the difference between the start and end $CH_4$ concentration using an enlarged MW with a minimum length of 7.5 minutes. To avoid measurement artefacts (e.g., overcompensation), being taken into account as start or end concentration, measurement points representing an inherent concentration change smaller or larger than the upper and lower quartile $\pm$ 0.25 times IQR were discarded prior to calculation of the total $CH_4$ flux. Third, the proportion of the total $CH_4$ emission released via ebullition was estimated following Eq. (2).

$$CH_{4_{ebullition_n}} = \sum_{i=1}^{n}(CH_{4_{total}} - CH_{4_{diffusion}}) \qquad (2)$$

Since no emergent macrophytes were present below the automatic chambers, plant-mediated transport of $CH_4$ was assumed to be zero. The same accounts for negative estimates of CH4 released through ebullition. To exclude measurement artefacts triggered by the process of closing the chamber, a death band of 25 % was applied to the beginning of each measurement prior to all flux calculation steps. The used R-script, a manual and test dataset are available at https://zenodo.org/record/53168.

### 2.3 Verification of applied flux separation algorithm

A laboratory experiment under reasonable controlled conditions was performed to verify the used flux separation algorithm. In order to artificially simulate ebullition events, distinct amounts (5, 10, 20, 30 and 50 ml) of a gaseous mixture (25 000 ppm $CH_4$ in artificial air; Linde, Germany) were inserted by a syringe through a pipe into a water filled tub covered with a closed chamber (V=0.114 m³; A= 0.145 m²). Airtight sealing was ensured by a water-filled frame connecting tub and chamber. The chamber was ventilated by a fan and connected via pipes to a Los Gatos Greenhouse Gas Analyser (911-0010, Los Gatos), measuring $CH_4$ concentrations inside the chamber with a 1 Hz frequency (Fig. 2). To ensure comparability

between *in vitro* and *in situ* measurements, data processing was performed based on 0.066 Hz records. The expected concentration changes within the chamber headspace as the result of methane injections was calculated as the mixing ratio between the amount of inserted gaseous mixture (25 000 ppm) and the air filled chamber volume (2 ppm).

## 3. Results and Discussion


The assumption of using sudden changes in chamber-based $CH_4$ concentration measurements to detect ebullition events was verified by the conducted laboratory experiment. Calculations of the simulated ebullition events and the amount of injected $CH_4$ showed a good overall agreement, which indicates the accuracy of the calculation algorithm (Fig. 4). However, flux

separation might be hampered due to a steady flux originating from other processes than diffusion through peat and water layers, such as the steady ebullition of micro bubbles (Prairie and del Giorgio 2013; Goodrich et al. 2011). A potential resulting impact on estimates of diffusive $CH_4$ emissions can be minimized by an enhanced frequency of concentration measurements during chamber closure. Based on a high temporal resolution, small-scale

differences within measured concentration changes can be identified and filtered by the variable IQR-criterion, which thereby reduces the detection limit of ebullition events. Compared to direct measurements of either diffusion or ebullition, as reported by e.g. Bastviken et al. (2010), the presented calculation algorithm prevents an interfering influence of spatial heterogeneity on separated ebullition and diffusion $CH_4$ fluxes, since both flux

components are derived during the same measurement and the same spatial entity. Moreover, the integration of the ebullition component into measurements rather than the calculation of single ebullition events ensure a reliable flux separation despite of potential measurement artefacts such as overcompensation or incomplete ebullition records (Goodrich et al. 2011; Miller and Oremland 1988). In case of a low water level, such as within the presented study

(<35cm) or parallel measurements of different trace gases (e.g., $CO_2$ and $CH_4$), the use of direct measurement systems for either ebullition (gas traps, funnels) or diffusion (bubble shields) might be limited. Hence, the presented simple and robust calculation algorithm, which is purely based on data processing, seems to be applicable to a broader range of different manual and automatic closed chamber systems, instrumental setups, study designs,

and ecosystems.

Due to the performed flux separation, the accuracy of temporal tendencies within the exemplary field data set was improved (Fig. 4, Tab. 1) and explanatory approaches could be



addressed. Total CH$_4$ emissions spatially integrated over the study period, as well as the respective contributions of ebullition and diffusion are shown in Fig. 5. Apart from short-term measurement gaps, a considerable loss of data occurred between the 27$^{th}$ of July and 7$^{th}$ of August 2013 due to malfunction of the measurement equipment.

In general, biochemical processes driving CH$_4$ production are closely related to temperature regimes (Christensen et al. 2005), determining the CH$_4$ production within the sediment (Bastviken et al. 2004). Hence, measured total CH$_4$ emissions showed distinct seasonal patterns following the temperature regime at 10 cm sediment depth. In addition to seasonality, total CH$_4$ emissions also featured diurnal dynamics, with lower fluxes during daytime and higher fluxes during nighttime, which were most pronounced during July and early September (Fig. 5). Especially during August, the diurnal variability was superimposed by short-term emission events and high amplitudes in recorded total CH$_4$ emissions. Similar to total CH$_4$ emissions, diffusive fluxes also showed a distinct temperature-driven seasonality as well as clear diurnal patterns throughout the entire study period (Fig. 6). However, compared to the diurnal variability of the total CH$_4$ fluxes, a pronounced shift of maximum CH$_4$ emissions from night- to daytime was revealed for the diffusive flux component (Fig. 5, Fig. 6). While maximum diffusive fluxes during July were recorded during nighttime hours (approx. 21:00 to 6:00), a shift to the daytime started in August, with maximum fluxes in September occurring between 0:00 and 11:00 (Fig. 6). This might be explained by differences in turbulent mixing due to changing water temperature gradients. During daytime, the surface water is warmed, thus preventing an exchange with the CH$_4$-enriched water near the sediment, which results in lower diffusive CH$_4$ emissions. During night time, when the upper water layer cools down and mixing is undisturbed, enhanced diffusive CH$_4$ emissions can be detected. This dynamics are more pronounced during warm days, explaining the seasonal shift, and concealed during periods with a high wind velocity. The obtained diurnal trend is in accordance with findings of Sahlée et al. (2014) and Lai et al. (2012), who reported higher night time and lower daytime CH$_4$ emissions for a lake site in Sweden and an ombrothropic bog in Canada, respectively. However, an opposing tendency was found by Deshmukh et al. (2014), who reported higher daytime and lower night time CH$_4$ emissions from a newly flooded subtropical freshwater hydroelectric reservoir within the Lam Theun river valley, Laos. In contrast to diurnal trends obtained for the total as well as diffusive CH$_4$ emissions, estimated ebullition events occurred erratically and showed neither clear seasonal nor diurnal dynamics. Nonetheless, periods characterized by more pronounced ebullition seemed to roughly follow the sediment temperature-driven CH$_4$ production within the sediment as e.g.



reported by Bastviken et al. 2004 (Fig. 5). This is confirmed by a distinct correlation between daily mean sediment temperatures and corresponding sums of measured ebullition fluxes. Moreover, fewer and smaller ebullition events were detected in times of reduced wind

velocity and high relative humidity (RH), for example from 10[th]-11[th] September and 18[th] -19[th] September 2013. However, at the level of single flux measurements, no significant dependency was found between the recorded environmental drivers and $CH_4$ release via ebullition. The relative contributions of diffusion and ebullition were 55 % (min. 33 % to max. 70 %) and 46 % (min. 30 % to max. 67 %), respectively. This is in accordance with

values reported by Bastviken et al. (2011), who compiled $CH_4$ emission estimates from 474 freshwater ecosystems with clearly defined emission pathways. A similar ratio was also found by Tokida et al. (2007), who investigated the role of decreasing atmospheric pressure as a trigger for $CH_4$ ebullition events in peatlands.

Comparison of flux data among the four chambers reveals considerable spatial heterogeneity

within the measured transect (*data not shown*). Monthly averages of diffusive, ebullative and total $CH_4$ emissions for all four chambers of the established transect as well as statistics showing the explanatory power of different environmental variables are summarized in Tab. 1. With respect to total $CH_4$ emissions, neighbouring chambers generally featured high differences in $CH_4$ fluxes, with no obvious trend along the transect. The same holds to be true

for derived ebullative and diffusive $CH_4$ flux components. After separation into diffusion and ebullition, dependencies with respect to different environmental drivers were revealed (Tab. 1).

## 4.  Conclusions

The results of the laboratory experiment as well as the estimated relative contributions of ebullition and diffusion during the field study indicate that the presented algorithm for $CH_4$ flux calculation and separation into diffusion and ebullition delivers reasonable and robust results. Temporal dynamics, spatial patterns and relations with environmental parameters well established in the scientific literature, such as sediment temperature, water temperature

gradients and wind velocity, became more pronounced when analysed separately for diffusive $CH_4$ emissions and ebullition. However, not all ebullition events (e.g., micro bubbles) seemed to be filtered correctly, as detected in case of enhanced $CH_4$ emissions during the beginning of August and the thereby superimposed diurnal cycling. Hence, further adaptation of measurement frequency and/or the applied data processing algorithm is required. In a next



step, the flux separation algorithm should be systematically tested against flux estimates generated with methods for measuring either ebullition or diffusion, such as bubble traps or bubble barriers. Moreover, the algorithm needs to be tested and evaluated with regards to generalizability and applicability to other freshwater and wetland ecosystems. Despite the mentioned shortcomings, the presented calculation approach for separating $CH_4$ emissions

increases the amount of information about the periodicity of $CH_4$ release and may help to reveal the influence of potential drivers as well as to explain temporal and spatial variability within both separated flux components.

**Acknowledgment**

This work was supported by the interdisciplinary research project CarboZALF, the Helmholtz Association of German Research Centres through a Helmholtz Young Investigators Group grant to T. S. (grant VH-NG-821), and infrastructure funding through the Terrestrial Environmental Observatories Network (TERENO). The authors want to express their special thanks to Mr. Marten Schmidt for construction as well as continuous maintenance of the auto-

chamber system and creative solutions for all kind of technical problems. The authors are also thankful to Mr. Bertram Gusovius for his kind help during performance of the lab experiment.







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





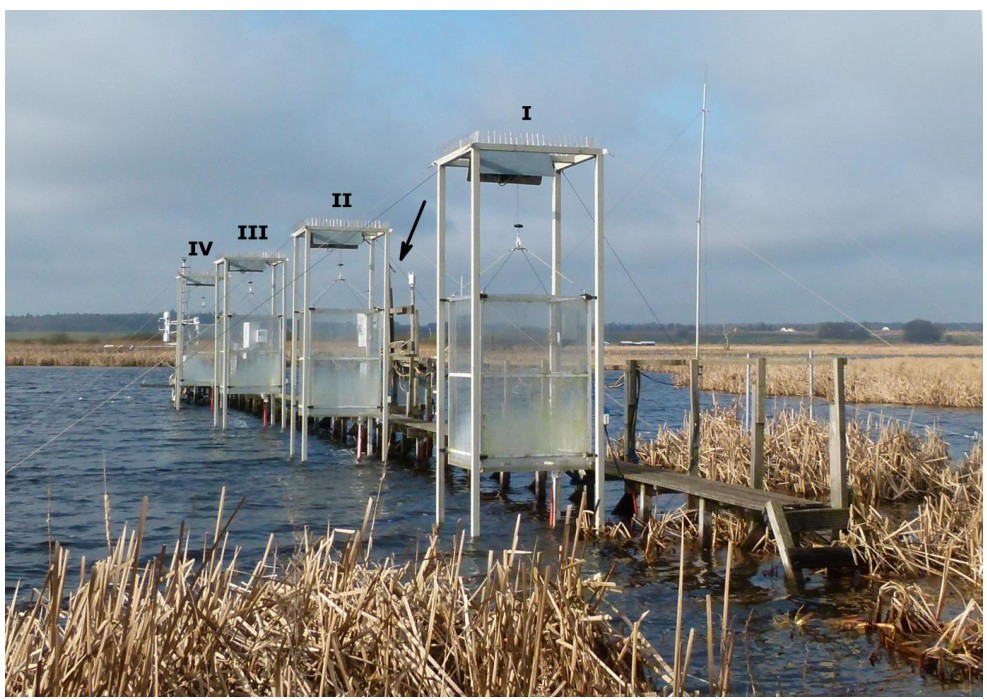

**Fig.1:** Transect of automatic chambers (AC) established at the measurement site. The arrow

indicates the position of the climate station near chamber II.







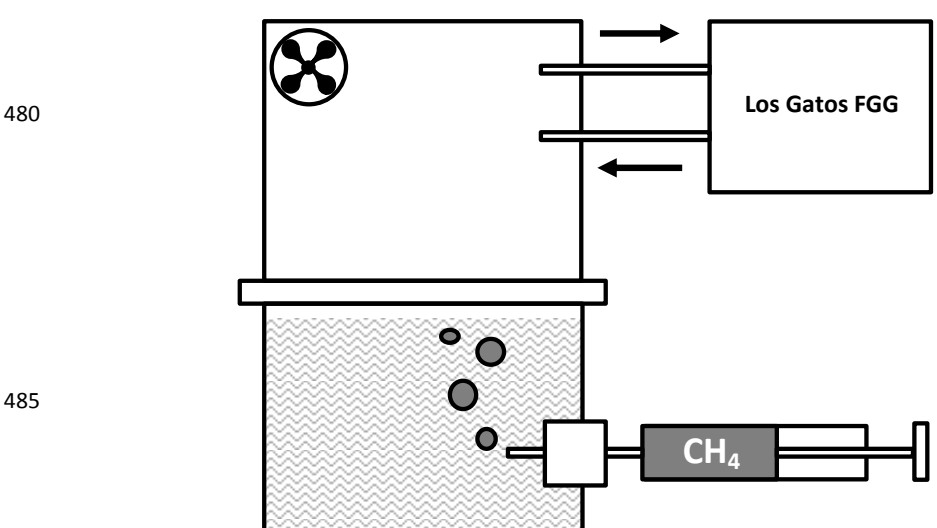

**Fig.2:** Scheme of experimental setup used for the simulation and determination of ebullition events with a Los Gatos Fast Greenhouse Gas (FGG) Analyser (911-0010, Los Gatos). The

crimped area represents water filled tub. Injections of gaseous mixture (25000 ppm $CH_4$ within artificial air; Linde, Germany) amounted to 5, 10, 20, 30 and 50 ml.





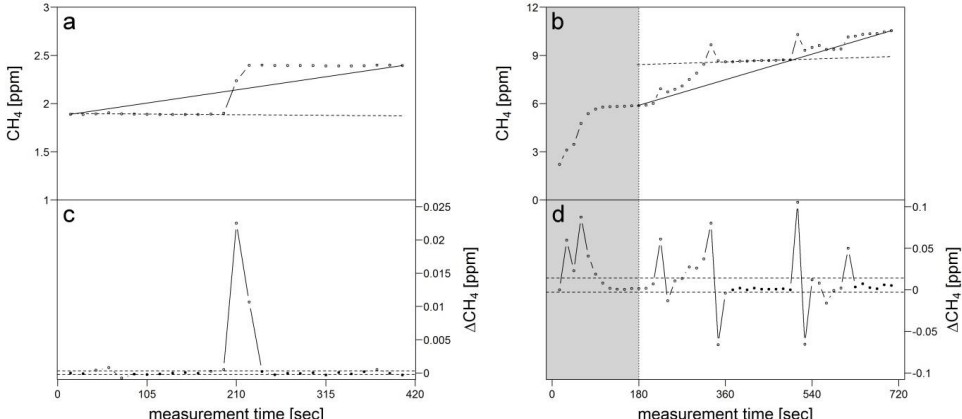

**Fig.3:** Scatterplots of recorded concentrations (ppm) within the chamber headspace for (a) a
simulated ebullition event and (b) an exemplary $CH_4$ measurement. The respective total $CH_4$
emission rate is represented by the black solid line, whereas $CH_4$ released by diffusion is
shown as a dashed line. The calculation of the corresponding diffusive flux is based on (c-d)
concentration changes (ppm) between measurement points. Time spans dominated by
diffusive $CH_4$ release are marked by black dots, enclosed by the 25 % and 75 % quantiles ±
0.25 IQR of obtained concentration changes, shown as black dashed lines. Unfilled dots
outside the dashed lines display ebullition events (see also Goodrich et al. 2011; Miller and
Oremland 1988). Gray shaded areas indicate the applied deathband (field study) at the
beginning of each measurement (25%).



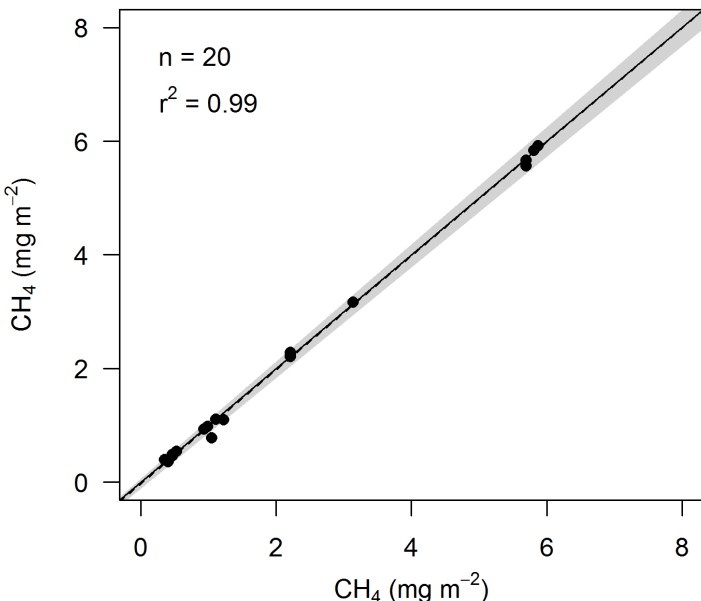

**Fig. 4:** Scatterplot of the amount of injected CH$_4$ and the corresponding calculated CH$_4$ ebullition event. The solid black line indicates the 1:1 agreement. The linear fit between the displayed values is represented by the black dashed line, surrounded by the 95% confidence interval (grey shaded area).




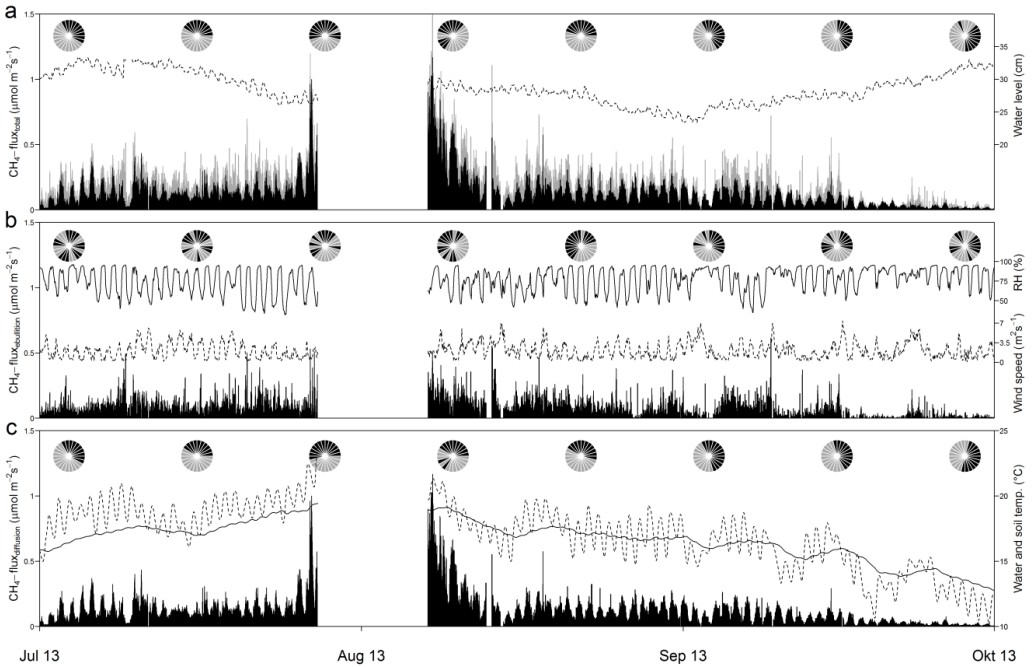

**Fig. 5:** Time series of (a) total $CH_4$ emissions and the corresponding amount of $CH_4$ released

via (b) ebullition and (c) diffusion during the study period from July until September 2013.

Development of important environmental parameters assumed to explain dynamics are also

shown ((a) water level, (b) RH and wind speed and (c) sediment (solid line) and water

temperature (dashed line)). Pie charts represent the biweekly pooled diurnal cycle of

measured $CH_4$ fluxes. Slices are applied clockwise, creating a 24-hour clock, with black and

light grey slices indicating hours with $CH_4$ flux above and below the daily mean, respectively.



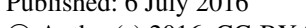

**Fig. 6:** Montly averaged diurnal cycle of diffusive CH$_4$ fluxes indicating differences in magnitude and amplitude as well as a shift in minimum and maximum daily CH$_4$ fluxes over the turn of the study period.



**Tab. 1:** Monthly averages ± 1 standard deviation of hourly $CH_4$ emissions (mg m$^{-2}$ h$^{-1}$) for the
chamber transect (from chamber I-IV, starting near the shoreline). Average standardized
(beta) coefficients and Nash-Sutcliff's-Efficiency (NSE) based on linear regressions and
multiple linear regressions between different environmental drivers and daily subsets of
calculated $CH_4$ emissions are shown below. Monthly averages as well as statistics are
separated according to diffusion, ebullition and total $CH_4$ flux. Superscript numbers indicate
significant differences between chambers. p-values of applied linear and multiple linear
regressions are indicated via asterisks.

| Month | Chamber | $CH_{4_{diffusion}}$ | $CH_{4_{ebullition}}$ | $CH_{4_{total}}$ |
|---|---|---|---|---|
| | | mg m$^{-2}$ h$^{-1}$ | | |
| July | I | 4.6[24] ± 3.1 | 5.5 ± 7.0 | 10.1[24] ± 7.8 |
| | II | 1.8[134] ± 1.5 | 3.7 ± 6.9 | 5.5[134] ± 7.1 |
| | III | 6.1[24] ± 4.0 | 4.7 ± 6.9 | 10.7[24] ± 8.2 |
| | IV | 8.7[123] ± 5.9 | 4.7 ± 5.3 | 13.3[123] ± 7.6 |
| August | I | 5.1 ± 5.9 | 5.0[24] ± 6.8 | 10.1 ± 10.0 |
| | II | 3.7 ± 5.0 | 2.9[14] ± 6.0 | 6.5 ± 8.6 |
| | III | 5.7 ± 4.9 | 5.8[24] ± 7.4 | 11.5 ± 9.5 |
| | IV | 6.1 ± 6.8 | 3.0[13] ± 5.0 | 9.1 ± 9.4 |
| September | I | 2.3[24] ± 2.0 | 1.8[24] ± 3.9 | 4.1[24] ± 4.8 |
| | II | 2.6[1] ± 2.7 | 1.1[13] ± 3.0 | 3.7[13] ± 4.4 |
| | III | 3.9[4] ± 3.9 | 5.4[24] ± 6.9 | 9.3[24] ± 8.8 |
| | IV | 1.3[13] ± 1.6 | 0.7[13] ± 3.4 | 2.1[13] ± 4.0 |
| Mean | | 5.1 ± 5.7 | 4.2 ± 6.5 | 9.2 ± 9.6 |
| Driver | | $CH_{4_{diffusion}}$ | $CH_{4_{ebullition}}$ | $CH_{4_{total}}$ |
| | | average standardized (beta) coefficient of daily data subsets | | |
| wind velocity | | -0.4˙ | -0.1 | -0.3˙ |
| relative humidity (RH) | | 0.5* | 0.1 | 0.4˙ |
| Air pressure | | 0.0 | -0.1 | 0.0 |
| water level | | -0.5* | -0.1 | -0.4˙ |
| air temp. (2 m) | | -0.6* | -0.1 | -0.4˙ |
| water temp. (5 cm) | | 0.1˙ | 0.1 | 0.1˙ |
| sediment temp. (2 cm) | | 0.3˙ | 0.0 | 0.2˙ |
| Δ water-air temp. | | 0.6* | 0.1 | 0.4˙ |
| average NSE of MLR | | 0.72 | 0.30 | 0.51 |

[1234] significant difference (α=0.1) between chamber I (1), II (2), III (3) and IV(4)
˙,* significant dependency with average p-value < 0.2 and p-value < 0.1

562