# Peer review of "A simple calculation algorithm to separate high-resolution $CH_4$ flux measurements into ebullition and diffusion-derived components"

_Atmospheric Measurement Techniques, 2016_

## Referee Comment (RC1) · Anonymous Referee #1 · 13 Jul 2016

**General Comments**

The authors present the validation and application of a data processing algorithm, which allows to distinguish and quantify erratic (ebullition) and diffusive CH4 emissions detected in an automatic chamber time series. This approach facilitates automatic data analysis of chamber measurements, which is an important step towards an improved comparability of these measurements. Secondly, it allows to identify erratic events of CH4 release from wetland ecosystems and to quantify the magnitude of

this release in comparison to merely diffusive emissions. This is important, as many studies have so far focused on diffusive emissions, which account only for a fraction of the total CH4 emissions. The authors thus provide a very useful tool that serves to advance the processing of data derived through automated chamber measurements. It is thus within the scope of AMT and of high interest to the scientific community. I find the study very interesting and useful, but I have a range of suggestions how the paper could be significantly improved.

According to the aims and scope of AMT, the journal's aim is to foster scientific discussion about advances in measurement techniques and data processing methods. Therefore, in my mind, the focus of an AMT paper should be to describe a method in a way that others can easily understand it and use it for their own research. With this in mind, I would like to suggest that the authors put more emphasis on explaining the method they use. The paper is a bit short in details about the data processing and the validation of the method and puts more focus on the results from a field campaign. Those are important, but it should be clear that the method is the main point and that the field study served as a test of the method's applicability and potential.

My second main suggestion is that the authors should include more details about the validation of the method and a more thorough discussion of the potential errors. The reader is, at the moment, unable to judge whether this automated data analysis will always lead to accurate results or when it might fail. Also, it is unclear whether this automated data analysis yields better or worse results than manual data analysis, whereas each measurement is looked at and fitted individually. It is also unclear how the algorithm handles data gaps, disturbances and artifacts. Would it work with other automated chamber systems as well or only with yours? Finally, plant-mediated transport is important. Does your algorithm only work at sites where plant-mediated transport can be excluded or is there potential to further develop the algorithm in the

future so that plant-mediated transport could be included?

**Specific Comments**

p. 2, line 43
The validation of the method in the lab should be mentioned in the Abstract.

p.3, line 67
I would mention here that CH4 is a greenhouse gas, which explains the relevance of CH4 flux measurements in the very beginning of the paper.

p. 5, line 140
I am wondering if the order of the subsections could be changed. The way it is structured, most emphasis is on the field measurements, whereas I have the feeling that the focus should be on the algorithm and its validation, and then on the field measurements. Not sure if it is possible to change the order though, because obviously information about the chamber system is needed before the algorithm can be introduced.

p. 5, line 153 and following
Do you have references for the composition of the vegetation? Has the study site been described previously?

p.6, line 165
The chamber system seems to be quite sophisticated. Is it a commercial system or did you develop it? Is it described elsewhere? If yes, please add references.

p.6, line 175
- Explain overcompensation or maybe show it in the Figure.
- Fig. 3 is referred to here, but Fig. 2 hasn't been mentioned yet. I suggest to adjust the order of the Figures.

p.6, line 179

[Figure]

Several questions remain open. 1) Over which time period were the measurements performed? This is mentioned in the abstract, but the information should be included here as well. 2) Were the chambers vented after each 10 min measurement? 3) How much time passed when the system switched to the next chamber? The reason for this question is the following: If you pump the analyzed air back into the chamber, you will have contamination every time you switch (i.e. air from chamber I is still in the analyzer, you switch to chamber II, air from chamber I will thus be returned to chamber II).

p. 6, line 181
- What was the water depth of the studied system?
- You say that temperatures were recorded at different (i.e. multiple?) water depths, but the only water depth you give is 5 cm above the sediment surface?

p. 6, line 191
I have a general question regarding the data analysis. Shouldn't you discard data after each ebullition event? The reason is the following: Let's say the chamber is closed and you have diffusive emissions in the beginning. They are driven by the gradient between water CH4 concentration and chamber CH4 concentration. After an ebullition event, the CH4 concentration in the chamber is enhanced over the normal boundary layer concentration, therefore, you will have reduced diffusive emission. Isn't that a systematic error? Can you estimate the magnitude of this error?

p. 6, line 192
Even though the script has been described elsewhere, I'd suggest you give a brief summary of the data processing nevertheless. Otherwise it will be hard for the reader to follow.

p.7, line 199
You should list which values were used for the user-defined parameters (maybe as a Table)

p. 7, line 207

I think it would be good to include a flow chart to support your explanation of how the algorithm works. It would make it easier to follow. In general, the description of how the algorithm works could be a bit more extensive and possibly be supported by graphics (e.g. flow chart, example data)

p.7, line 216
"To exclude measurement artifacts triggered by the process of closing..." This information should appear earlier in the Section, you should describe first which data is discarded and then how fluxes are derived from the remaining data.

p. 7, line 222
This is a nice way to validate the algorithm for ebullition events. Was the algorithm also somehow verified for the diffusive flux? Maybe previously? This would be an important information.

p.8, line 235
At present, the Results and Discussion Section is not very well structured and it is easy to mix up the different experiments. It has to be made clear that what you did was a two-step approach: First you validated the algorithm by testing it under lab conditions, second you applied the algorithm to field data. The reader could be under the impression that you're validating your method with field data, but of course it is the lab measurements that support your theory. The field data is to show how useful your algorithm is for the quantification and interpretation of fluxes. Therefore, I would like to suggest to structure the Results and Discussion Section into 3.1 Validation of the algorithm through laboratory measurements, 3.2 Application of the algorithm to field data, 3.3 Overall performance of the algorithm. That would help the reader distinguish the different experiments. I think that the lab measurements need more discussion – it is the evidence that your method works for ebullition events. But you should also discuss potential errors. In 3.3 you could evaluate the overall performance of the algorithm, the advantages it has, but also include a discussion of potential errors.

p.9, lines 267-270
Your reasoning is: In the literature, it has been shown that CH4 production is related to temperature. Therefore, our measurements show a pattern that relates CH4 to temperature. But actually the reasoning is the other way around: You find in your data that CH4 is related to temperature. This is in accordance with the literature.

p.9 lines 282-286
You do have the data to support this theory (you mention that you measured the water temperature at different depths). I suggest to use your data to prove your theory.

p. 10, line 297
What exactly is the correlation between temperature and ebullition fluxes? I'd suggest to either give a correlation coefficient here or to include a Figure.

p.10, line 308
Does the contribution of ebullition to the total flux (in %) also exhibit a diurnal pattern?

p.10, line 317
In what I suggested to be Section 3.3, I would recommend that you also include a short outlook as to which further developments the algorithm requires and what its potential is to be used as a general tool for automated chamber measurements (kind of what you're doing in your conclusions). Do you think it is possible to integrate plant-mediated fluxes in the future or is your algorithm only applicable in systems where these can be neglected? You should also answer the question under which circumstances the performance of the algorithm might be poor, and which errors can be expected. Could just anyone who measured a chamber time series use your algorithm and get reliable results? Do the flux estimates derived with your algorithm have a robust error propagation estimation?

Fig. 2
This Figure would benefit from annotations (e.g. the fan, the chamber, water tub). "Injections of gaseous mixture amounted to ..." - this information is not relevant in the

caption and is already given in the text.

Fig. 3

This Figure is not very readable and very complex. To make it easier for the reader to understand the Figure, I suggest the following changes: Data points should be bigger, it is almost impossible to distinguish open and black circles. Axis labels should be bigger. The Figure needs a legend that allows the reader to see what the dashed/solid lines and open/black circles denote without having to read the caption. At the same time, if this legend is included, you can remove the extensive and somewhat complicated descriptions of dashed/solid lines, open/black circles in the caption. Why was no death band applied in a and c?

Fig. 4

The data points should have error bars. The axis limit could be reduced to 7. If $r^2$ is shown, I'd suggest to also show p and the calculated slope and intercept of the regression line.

Fig. 5

This is a very interesting way to present your data. However, similar to Fig. 3, the Figure is very complex and not easily readable. I would like to suggest bigger labels, and a legend like I said in my comment above. A general question, does the bottom slice of the pie (i.e. 6 o'clock on a normal clock) correspond to 12 o'clock noon? If this is correct, then maybe it is good to warn the reader that what he normally perceives as 6 o'clock is not 6 o'clock in this Figure. I think it would be a good idea to have an "example clock-pie" with the actual hours (Let's say, 0:00, 6:00, 12:00, 18:00 ) next to the Figure so that it is easier to understand the clock-concept at first glance, otherwise the clock-concept might be a bit misleading.

**Technical corrections**

p. 2, line 46
change to "given in the literature"
[Figure]

p.3, line 77
What does "at all scales" refer to?

p.5, line 148
change to "in the beginning"

p.5, line 150
change to "were reported"

p.5, line 156
"below the chambers"

p.7, line 203
"outlier"

p. 7, line 222
I am unsure about "reasonable controlled conditions". I'd suggest to delete "reasonable"?

p. 8, line 232
change to "were calculated"

p. 8, line 262
"explanatory approaches could be addressed" - I think the wording needs to be changed here.

p. 9, line 286
"This dynamics are ..." should be changed to "These dynamics are"

p. 9, line 291
daytime and nighttime are sometimes written as day time and night time throughout the text (here it is just most obvious because there are two different versions in the same sentence). Please check the article for consistent spelling of those terms.

p.13, line 393

I think the title of that publication should be "Automated modeling of ecosystem CO2 fluxes based on periodic closed chamber measurements: ..."
* * *

---

## Author Comment (AC1) · 28 Oct 2016

**General Comments**

1. I find the study very interesting and useful, but I have a range of suggestions how the paper could be significantly improved. According to the aims and scope of AMT, the journal's aim is to foster scientific discussion about advances in measurement techniques and data processing methods. Therefore, in my mind, the focus of an AMT paper should be to describe a method in a way that others can easily understand it and use it for their own research. With this in mind, I would like to suggest that the authors put more emphasis on explaining the method they use. The paper is a bit short in details about the data processing and the validation of the method and puts more focus on the results from a field campaign. Those are important, but it should be clear that the method is the main point and that the field study served as a test of the method's applicability and potential.

We agree and restructured and rewrote the entire section 2.2 which explains the flux calculation and separation approach in more detail now. To further support the more detailed explanation a flow chart was added to the MS as well (please see changed section 2.2 and flow chart below).

*" $CH_4$ flux calculation and separation was performed based on an adaptation of a standardized R-script presented in detail by Hoffmann et al. (2015). Fig. 2 shows a flow chart of the flux calculation algorithm implemented in R and the principle of the performed $CH_4$ flux separation. To estimate the relative contribution of diffusion and ebullition to total $CH_4$ emissions, flux calculation was performed twice, once for the total $CH_4$ flux ($CH_{4_{total}}$) and once for the diffusive component of the flux rate ($CH_{4_{diffusion}}$), by adjusting selected user-defined parameter setups of the used R-script (Hoffmann et al. 2015). Prior to each flux calculation a death band of 25 % (user defined) was applied to the beginning of each flux measurement, thus excluding measurement artefacts triggered by the process of closing the chamber. On the remaining flux measurement data a variable moving window (MW) with a minimum size of 5 ($CH_{4_{diffusion}}$; user defined) and 30 consecutive data points ($CH_{4_{total}}$; user defined) was applied, generating several data subsets per flux measurement for $CH_{4_{diffusion}}$ and one data subset for $CH_{4_{total}}$. Subsequently, $CH_4$ fluxes were calculated for all data subsets per flux measurement using Eq. (1), where M is the molar mass of $CH_4$, δv is the linear concentration change over time (t), A and V denote the basal area and chamber volume, respectively, and T and P represent the inside air temperature and air pressure. R is a constant (8.3143 $m^3$ $Pa$ $K^{-1}$ $mol^{-1}$). In case of $CH_{4_{total}}$ the difference between the start and end $CH_4$ concentration of the enlarged MW (7.5 minutes) was used (δv) instead of the slope of the linear regression fit ($CH_{4_{diffusion}}$). To avoid measurement artefacts (e.g., overcompensation), being taken into account as start or end concentration, measurement points representing an inherent concentration change smaller or larger than the upper and lower quartile ± 0.25 times IQR (user defined) were discarded prior to calculation of $CH_{4_{total}}$. In case of diffusion the resulting numerous $CH_4$ fluxes calculated per measurement (based on the moving window data subsets) were further evaluated according to different exclusion criteria: (i) range of within-chamber air temperature not larger than ± 1.5 K; (ii) significant regression slope (p ≤ 0.1, t-test); and (iii) non-significant tests (p > 0.1) for normality (Lillifor´s adaption of the Kolmogorov-Smirnov test), homoscedasticity (Breusch-Pagan test) and linearity. In addition (iv) abrupt concentration changes within each MW data subset were identified by a rigid outlier test, which discarded fluxes with an inherent concentration change outside of the range between the upper and lower quartile ± 0.25 times (user defined) the interquartile range (IQR). Calculated $CH_{4_{diffusion}}$ fluxes which did not meet all exclusion criteria were discarded. In case of more than one flux per measurement meting all exclusion criteria, the $CH_{4_{diffusion}}$ flux with a starting $CH_4$ concentration being closest to the atmospheric $CH_4$ concentration was chosen. Finally, the*

*proportion of the total $CH_4$ emission released via ebullition was estimated by subtracting identified $CH_{4_{diffusion}}$ from the calculated $CH_{4_{total}}$ following Eq. (2).*

$$CH_{4_{ebullition_n}} = \sum_{i=1}^{n}(CH_{4_{total}} - CH_{4_{diffusion}}) \qquad\qquad (2)$$

*Since no emergent macrophytes were present below the automatic chambers, plant-mediated transport of $CH_4$ was assumed to be zero. The same accounts for negative estimates of CH4 released through ebullition. The used R-script, a manual and test dataset are available at https://zenodo.org/record/53168.".*

[Figure]

2. My second main suggestion is that the authors should include more details about the validation of the method and a more thorough discussion of the potential errors. The reader is, at the moment, unable to judge whether this automated data analysis will always lead to accurate results or when it might fail.

We agree and restructured and rewrote section 3 according to the suggestions made by both reviewers. The section was divided into the subsection 3.1 "Verification of the flux separation algorithm" (including results of the performed laboratory experiment and their discussion), 3.2 "Application to an exemplary field study" (including results and their discussion gained through field measurements) and 3.3 "Overall performance" (including a more thoroughly discussion of advantages and limitations/potential error sources of the presented flux separation algorithm). In addition we refer to eddy covariance measurements made at the same study site during the same period (Franz et al. (2015)), which resulted in total $CH_4$ fluxes being comparable in magnitude and seasonal development to fluxes measured by the chamber system and calculated by the presented data processing algorithm.

*"$CH_{4_{total}}$ fluxes observed by the AC system and calculated with the presented algorithm were comparable to $CH_4$ emissions measured during the study period by a nearby eddy covariance system (Franz et al. 2015)."*

3. Also, it is unclear whether this automated data analysis yields better or worse results than manual data analysis, whereas each measurement is looked at and fitted individually.
   We think that automatization and standardization is a perquisite for scientific studies, helping to produce traceable, reproducible and comparable results/data. This is most often hard to achieve by manual data analysis which is not only time consuming (the presented data set contains more than 14.000 flux measurements), but also somehow subjective (due to individual expert knowledge and decision making). During recent studies, we compared manually calculated $CO_2$-fluxes with an automatic and standardized calculation tool using R (Hoffmann et al. 2015). Whereas the automatic calculation did not differ between multiple calculations using the same user defined parameter setups and data sets, the manual calculations made by different researchers were characterized by a rather high variability. The same accounts for repetitive calculations of the same data set by one researcher.

4. It is also unclear how the algorithm handles data gaps, disturbances and artifacts.
   The algorithm requires consecutive concentration records during single chamber measurement. In case of missing measurements (such as during end of July and beginning of August) no gap filling was performed. Based on revealed flux component specific temporal dynamics and dependencies, empirical modelling might be included into the algorithm. In case of measurement gaps within one single measurement, the measurement is divided by the algorithm into two measurements (one before and one after the gap). However, such data gaps did not occur during this study. If the divided measurement are too short for the algorithm (smaller than the minimum moving window size) they are discarded. A number of filters try to discard disturbances and artifacts from calculated fluxes. To better address this issue we rewrote and extended section 2.2 (please see answer to 1. comment reviewer #1).

5. Would it work with other automated chamber systems as well or only with yours?
   In principle the algorithm is supposed to work with all kind of automatic or manual closed chamber systems, as long as they deliver consecutive records for $CH_4$ concentrations and air temperature. This means that the only requirement is a proper data format (csv-file), which is used by the R-algorithm as source data to calculate/separate the CH4 measurements into its components.

6. Finally, plant-mediated transport is important. Does your algorithm only work at sites where plant-mediated transport can be excluded or is there potential to further develop the algorithm in the future so that plant-mediated transport could be included?
   To date, the presented algorithm separates only diffusion and ebullition, and will thus only work on open-water systems were the plant-mediated transport can be neglected. We, however, are

working on an approach to also include plant mediated transport into the calculation and separation algorithm. Therefore, below surface (sediment/water) measurement devices to record the $CH_4$ concentration are needed. Based on this, diffusive $CH_4$ emissions could be calculated via a gradient approach, while the chamber still measures the total $CH_4$ flux. By subtracting the diffusive from the total flux the sum of plant-mediated and ebullition flux could theoretically be derived. Both flux components could be subsequently separated, using the presented approach, assuming plant-mediated transport being more continuous than ebullition. This however is not implemented yet, since we still work on the measurement device. The advantage however, is clear since we would be thus able to measure all three flux components on the same measurement plot, without a bias due to spatial heterogeneity.

**Specific Comments**

7. p. 2, line 43 The validation of the method in the lab should be mentioned in the Abstract.
We agree and added the validation using the laboratory experiment to the abstract.

*"The algorithm was validated by performing a laboratory experiment and tested using flux measurement data (July to September 2013) from a former fen grassland site, which converted into a shallow lake as a result of rewetting."*

8. p.3, line 67 I would mention here that CH4 is a greenhouse gas, which explains the relevance of CH4 flux measurements in the very beginning of the paper.
We agree and changed the first sentence.

*"Wetlands and freshwaters are among the main sources for methane ($CH_4$), which is one of the major greenhouse gases (Dengel et al. 2013; Bastviken et al. 2011; IPCC 2013)."*

9. p. 5, line 140 I am wondering if the order of the subsections could be changed. The way it is structured, most emphasis is on the field measurements, whereas I have the feeling that the focus should be on the algorithm and its validation, and then on the field measurements. Not sure if it is possible to change the order though, because obviously information about the chamber system is needed before the algorithm can be introduced.
We agree. To better emphasize the calculation algorithm and its validations using the laboratory experiment and field study respectively, we changed the order of subsections.

      2.1 Automatic chamber system
      2.2 Flux calculation and separation algorithm
      2.3 Verification of applied flux separation algorithm
      2.4 Exemplary field study

Since a closed chamber system is the general basis for the calculation algorithm, details of the used automatic chamber system are mentioned at the beginning of the Material and Method section. After this, the flux calculation and separation algorithm as the actual focus of the paper is now mentioned, followed by its validation/verification in the lab. The subsection *"Ancillary field measurements"* was merged with *"Study site"* to the more general heading *"2.4 Exemplary field study"*.

10. p. 5, line 153 and following. Do you have references for the composition of the vegetation? Has the study site been described previously?
Details given about the vegetation are based on a monitoring performed at the study site during our project, wherefore no reference was given in the MS. However, the Study site as well as the composition of the vegetation was also described by Franz et al. (2015; Biogeosciences, doi:

10.5194/bg-13-3051-2016), Steffenhagen et al. (2012) and Hahn-Schöfl et al. (2011). We therefore added these references to the MS as well.

11. p.6, line 165 The chamber system seems to be quite sophisticated. Is it a commercial system or did you develop it? Is it described elsewhere? If yes, please add references.
The chamber system is a non-commercial system, developed by our working group. A comparable system was used for $CO_2$ measurements on an agricultural landscape within the study of Hoffmann et al. (2016; Biogeosciences Discuss., doi:10.5194/bg-2016-332, 2016). This study was not referred to, due to differences not only in the measured trace gas, but also regarding the used analyzer and measured ecosystem. Moreover, the article is only published as a discussion paper yet.

12. p.6, line 175 Explain overcompensation or maybe show it in the Figure.
We added a more detailed explanation for overcompensation to section 2.1.

*"However, due to the large chamber volume, complete mixture of the chamber headspace took up to 30 seconds. As a result of this, most peaks due to ebullition events were directly followed by a smaller decrease in measured $CH_4$ concentration. This signal indicates a short term overestimation of the ebullition event (peak), which was compensated after the chamber headspace, was mixed proper (decrease), a phenomenon further on referred to as overcompensation (Fig. 3)."*

In addition we now refer to overcompensation within the figure caption to Fig. 3.

*"Negative $\Delta CH_4$ values indicate an overcompensation due to (temporally) insufficient headspace mixing."*

13. Fig. 3 is referred to here, but Fig. 2 hasn't been mentioned yet. I suggest to adjust the order of the Figures.
We corrected the order of Figures throughout the MS.

14. p.6, line 179 Several questions remain open.
1) Over which time period were the measurements performed? This is mentioned in the abstract, but the information should be included here as well.
We included this information now as well in section 2.4.

*"Ecosystem $CH_4$ exchange was measured from beginning of July to end of September 2013 at a flooded former fen grassland site, located within the Peene river valley in Mecklenburg-Western Pomerania, northeast Germany (53°52′N, 12°52′E)."*

2) Were the chambers vented after each 10 min measurement?
Yes, the chambers were vented (using the internal fan) during the entire 50 min between two measurements of the same chamber. To better address this important information we added the following sentence to *"2.1 Automatic chamber system"*:

*"Each chamber was vented using the internal fan throughout the entire 50 min between two measurements at the same chamber position."*

3) How much time passed when the system switched to the next chamber? The reason for this question is the following: If you pump the analyzed air back into the chamber, you will have contamination every time you switch (i.e. air from chamber I is still in the analyzer, you switch to chamber II, air from chamber I will thus be returned to chamber II).

We think this is a really important issue and are glad for this valuable remark. In general, the tubes connected to the sensor were vented for 3 minutes between two measurements:

- 1 minute before switching to the next chamber using the air of the reopened chamber which was measured for 10 minutes.
- 2 minutes after switching to the next chamber, using air of the open chamber, to be measured next. After this 2 minutes the chamber will be automatically deployed on the frame.

Since the 1 minute before switching will be biased by the measured performed directly before, the tubes were vented using unbiased air for 2 minutes. We therefore added the following sentence to *"2.1 Automatic chamber system"*:

*"When switching from one chamber to another, the tubes were vented for two minutes using the air of the non-deployed chamber to be measured next."*

15. p. 6, line 181 What was the water depth of the studied system?
The water depth is shown in Fig. 5 (now 6), and ranged from 22 to 35 cm throughout the study period. In addition we added the limits to *"3.3 Overall performance"* as follows:

*"This is in particular the case, when measuring at wetland ecosystem with a varying water level, such as at the exemplary study site (22 to 35 cm)."*

16. You say that temperatures were recorded at different (i.e. multiple?) water depths, but the only water depth you give is 5 cm above the sediment surface?
We stated that we measured temperature in different water and soil depths *("Temperatures were recorded in different water (5 cm above sediment surface) and sediment depths (2 cm, 5 cm, and 10 cm below the sediment-water interface), using thermocouples (T107, Campbell Scientific).")*. This seems to be misleading, since we actually meant that we measured temperatures in four depths underneath the chambers (one water depth (5 cm) and three different sediment depths). We therefore changed the sentence to:

*"Temperatures were recorded in the water (5 cm above sediment surface) and different sediment depths (2 cm, 5 cm, and 10 cm below the sediment-water interface), using thermocouples (T107, Campbell Scientific)."*.

17. p. 6, line 191
I have a general question regarding the data analysis. Shouldn't you discard data after each ebullition event? The reason is the following: Let's say the chamber is closed and you have diffusive emissions in the beginning. They are driven by the gradient between water CH4 concentration and chamber CH4 concentration. After an ebullition event, the CH4 concentration in the chamber is enhanced over the normal boundary layer concentration, therefore, you will have reduced diffusive emission. Isn't that a systematic error?
In general, every new concentration record during one chamber measurements will be influenced by the concentration records before, as long as these records alter the concentration gradient. This is a well-known (e.g. Hoffmann et al. 2015), general limitation of closed chamber measurement system, and most often handled by reducing the measurement length or enhancing the chamber volume, which minimize the change within the concentration gradient. Regarding the specific measurement site we assume that the influence, however, will be negligible. The reason therefore is the rather high $CH_4$ concentration within the sediment as measured by Hahn-Schöfl et al. (2011). They reported $CH_4$ concentrations within organic sediment probes taken at the same site in 2007 for an incubation experiment ranging up to 500.000 ppm. As a result, even during bigger ebullition events, which might enhance the chamber air $CH_4$-Concentration up to 30 ppm, the actual gradient between sediment and atmospheric $CH_4$-concentration remains high. However, the

diffusion from sediment to water to atmosphere might be delayed, resulting in reduced diffusive CH$_4$ emissions after bigger ebullition events
.

18.
The magnitude of this error can be estimated by calculating the diffusive flux twice, using the presented tool and a variation within the user defined parameters: once disregarding the last e.g. 50% of each measurement, and once disregarding the first e.g. 50% of each measurement during the calculation. By comparing the resulting fluxes of both calculations, a systematic error and its magnitude would occur in terms of significantly lower fluxes during the second calculation. However, a significant difference between these two calculations was not obtained for the presented data set.

19. p. 6, line 192 Even though the script has been described elsewhere, I'd suggest you give a brief summary of the data processing nevertheless. Otherwise it will be hard for the reader to follow.
We rewrote section 2.2 and added a flow chart (see flow chart above) to further explain the developed algorithm

20. p.7, line 199 You should list which values were used for the user-defined parameters (maybe as a Table)
We agree and added user defined parameters and used values to section 2.2 of the MS and to flow-chart respectively.

21. p. 7, line 207 I think it would be good to include a flow chart to support your explanation of how the algorithm works. It would make it easier to follow. In general, the description of how the algorithm works could be a bit more extensive and possibly be supported by graphics (e.g. flow chart, example data)
We added a flow-chart to the MS (please see answer to 1. Comment reviewer #1) explaining how the flux calculation and separation algorithm is working.

22. p.7, line 216
"To exclude measurement artifacts triggered by the process of closing..." This information should appear earlier in the Section, you should describe first which data is discarded and then how fluxes are derived from the remaining data.
We rewrote section 2.2. As a result the information about discarded concentration records is appearing earlier now.

23. p. 7, line 222
This is a nice way to validate the algorithm for ebullition events. Was the algorithm also somehow verified for the diffusive flux? Maybe previously? This would be an important information.
No direct verification of the presented algorithm to calculate diffusive CH$_4$ emissions was made. However, the total CH$_4$-flux measured at the exemplary study site was verified (see figure below) by measurements of a nearby (>10 m distance to chambers) eddy covariance system performed during the same period. Both measurement devices (eddy covariance and automatic chambers) yielded in comparable results regarding the dynamics and magnitude of obtained total CH$_4$-emissions. Thus, ebullition (lab experiment) and total CH$_4$-emission (eddy covariance measurements) were verified. To better address this issue we added the following sentence to the MS:

[Figure]

*"$CH_{4total}$ fluxes observed by the AC system and calculated with the presented algorithm were comparable to $CH_4$ emissions measured during the study period by a nearby eddy covariance system (Franz et al. 2015)."*

**24. p.8, line 235**

At present, the Results and Discussion Section is not very well structured and it is easy to mix up the different experiments. It has to be made clear that what you did was a two-step approach: First you validated the algorithm by testing it under lab conditions, second you applied the algorithm to field data. The reader could be under the impression that you're validating your method with field data, but of course it is the lab measurements that support your theory. The field data is to show how useful your algorithm is for the quantification and interpretation of fluxes. Therefore, I would like to suggest to structure the Results and Discussion Section into 3.1 Validation of the algorithm through laboratory measurements, 3.2 Application of the algorithm to field data, 3.3 Overall performance of the algorithm. That would help the reader distinguish the different experiments.

We restructured and rewrote section 3. and subdivided it into:

3.1 Verification of the flux separation algorithm

3.2 Application to an exemplary field study

3.3 Overall performance

**25.** I think that the lab measurements need more discussion – it is the evidence that your method works for ebullition events. But you should also discuss potential errors.

We agree and extended this section, adding potential error sources which might occur during the lab experiments, but also problems which might occur when applying the algorithm under field conditions.

**26.** In 3.3 you could evaluate the overall performance of the algorithm, the advantages it has, but also include a discussion of potential errors. In what I suggested to be Section 3.3, I would recommend that you also include a short outlook as to which further developments the algorithm requires and what its potential is to be used as a general tool for automated chamber measurements (kind of what you're doing in your conclusions).

We extended the discussion by mention advantages compared to other direct or indirect flux separation approaches *("3.3 Overall performances")*. Furthermore, disadvantages/potential error sources of the presented algorithm are more thoroughly discussed now *("3.3 Overall performance")*.

*"Compared to direct measurements of diffusion or ebullition, as reported by e.g. Bastviken et al. (2010), the presented calculation algorithm in combination with the used AC system, features two major advantages. On the one hand it allows deriving the ebullition and diffusion flux components based on the same measurement and for the same spatial entity, which prevents an interfering influence of spatial heterogeneity on observed flux components. This is not the case for flux separation based on a combination of different measurement devices, such as automatic chambers and bubble traps, which need a sufficient number of repetitions and degree in data aggregation to reduce the bias, emerging from the spatiotemporal heterogeneity of erratically occurring ebullition events. On the other hand, the solely data processing based flux separations approach allows for an application , when the use of direct measurement systems for either ebullition (gas traps, funnels) or diffusion (bubble shields) might be limited. This is in particular the case, when measuring at wetland ecosystem with a varying water level, such as at the exemplary study site (22 to 35 cm). During the summer month 2009 and 2016 the water level dropped substantially, being either next to or even below the surface (data not shown). This limited the theoretical use of bubble traps and shields, despite of potential ebullition from the water saturated sediment, to periods with a sufficient water level, resulting in larger measurement gaps. Similar to that, parallel measurements of different trace gases (e.g., $CO_2$ and $CH_4$) are not affected by the presented flux separation algorithm.*

*However, flux separation using the presented algorithm might be biased by steady ebullition of micro bubbles and frequently occurring strong ebullition events. Steady ebullition of micro bubbles, results in an overestimation of $CH_{4_{diffusion}}$ and underestimation of $CH_{4_{ebullition}}$, an effect, which might be reduced by enhancing the measurement frequency and thus the sensitivity of the variable IQR-filter. Frequently occurring strong ebullition events, however, might disable the calculation of $CH_{4_{diffusion}}$, which hampers flux separation for the corresponding measurement. Out of 14.828 valid automatic chamber measurements, the algorithm failed to calculate $CH_{4_{diffusion}}$ during 170 measurements. This equals 1.15 % of all measurements. Taken into account that the presented measurement site is characterized by rather large $CH_4$ emissions (Franz et al. 2015) and frequently occurring ebullition events, this limitation seems to be negligible.*

*Compared to other data processing based approaches for $CH_4$ flux separation (e.g. Goodrich et al. 2011; Miller and Oremland 1988), the integration of the ebullition component into measurements rather than the calculation of single ebullition events ensure a reliable flux separation, despite of potential measurement artefacts such as overcompensation or incomplete ebullition records. As a result of this, the presented, data processing based approach will be applicable as long as the underlying closed chamber measurements deliver continuous data sets for $CH_4$ concentration and air temperature.*

*Accounting for the few prerequisites (high resolution closed chamber measurements) as well as mentioned advantages, an application of the presented approach to open-water areas of a broad range of wetland ecosystems is stated."*

27. Do you think it is possible to integrate plant-mediated fluxes in the future or is your algorithm only applicable in systems where these can be neglected?

As stated in the MS, to date the algorithm is only applicable to open water areas of wetland ecosystems, were plant-mediated transport can be neglected. However, as mentioned above (general comments), we added an outlook to section "4. Conclusions", which states that by using additional measurement devices to obtain $CH_4$ concentrations in the sediment and overlying water column, an extension of the presented algorithm to also include a separation of plant-mediated transport, might be theoretically possible.

*"During future studies, the possibility to implement the separation of $CH_4$ released through plant mediated transport into the presented algorithm should be addressed. This might be possible by*

*complete chamber CH₄ concentration measurements with CH₄ concentrations measured in different water and/or sediment depth. This might allow to directly deriving $CH_{4_{diffusion}}$ fluxes, whereas the remaining plant mediated transport and ebullition flux components could be separated using the algorithm."*

28. You should also answer the question under which circumstances the performance of the algorithm might be poor, and which errors can be expected. Could just anyone who measured a chamber time series use your algorithm and get reliable results?

We added sentences regarding the reliability of the presented approach to section 3.1 (micro bubbles) and 3.3 (frequent strong ebullition events). In principle the algorithm is supposed to work with all kind of automatic or manual closed chamber systems, as long as these systems deliver consecutive records for CH₄ concentrations and air temperature. The reliability however, is not only a question of the presented flux calculation and separation algorithm, but also of the underlying measurement data. The quality of the data depends on the used chamber designs (e.g. airtight sealing, ventilated, pressure equilibration), measurement settings (e.g. frequency of concentration records) as well as the measured ecosystem (does steady ebullition through micro bubbles occur?). To better address this important issue we added the following to *"3.3 Overall performance"*.

*"As a result of this, the presented, data processing based approach will be applicable as long as the underlying closed chamber measurements deliver continuous data sets for CH₄ concentration and air temperature.*
*Accounting for the few prerequisites (high resolution closed chamber measurements) as well as mentioned advantages, an application of the presented approach to open-water areas of a broad range of wetland ecosystems and closed chamber systems is stated."*

29. Do the flux estimates derived with your algorithm have a robust error propagation estimation?

Errors can be only calculated for the diffusive flux component, based on the underlying linear regression fit. Since the total flux is calculated using the difference between the start and end CH₄ concentration, an error estimation for single measurements/fluxes is not possible. This also prevents an error calculation for the ebullition flux component of a measurement. Thus errors are not given for all flux components of single measurements.

30. p.9, lines 267-270 Your reasoning is: In the literature, it has been shown that CH4 production is related to temperature. Therefore, our measurements show a pattern that relates CH4 to temperature. But actually the reasoning is the other way around: You find in your data that CH4 is related to temperature. This is in accordance with the literature.

We agree and rewrote this paragraph.

*"Measured total CH₄ emissions showed distinct seasonal patterns following the temperature regime at 10 cm sediment depth. This is in accordance with Christensen et al. (2005) and Bastviken et al. (2004), who showed that biochemical processes driving CH₄ production are closely related to temperature regimes, which determine the CH₄ production within the sediment."*

31. p.9 lines 282-286 You do have the data to support this theory (you mention that you measured the water temperature at different depths). I suggest to use your data to prove your theory.

We stated that we measured temperature in different water and soil depths *("Temperatures were recorded in different water (5 cm above sediment surface) and sediment depths (2 cm, 5 cm, and 10 cm below the sediment-water interface), using thermocouples (T107, Campbell Scientific).").* This seems to be misleading, since we actually meant that we measured temperatures in four depths underneath the chambers (one water depth (5 cm) and three different sediment depths). We therefore changed the sentence to:

*"Temperatures were recorded in the water (5 cm above sediment surface) and different sediment depths (2 cm, 5 cm, and 10 cm below the sediment-water interface), using thermocouples (T107, Campbell Scientific)."*.

32. p. 10, line 297 What exactly is the correlation between temperature and ebullition fluxes? I'd suggest to either give a correlation coefficient here or to include a Figure.
We agree and added the coefficients of determination to the MS.

*"This is confirmed by a distinct correlation between daily mean sediment temperatures and corresponding sums of measured ebullition fluxes ($r^2$: 2 cm = 0.63; 5 cm = 0.63; 10 cm =0.62)."*

The figures show the distinct correlation between the average daily sediment temperature (2 cm depth) and daily sums of calculated ebullition fluxes (upper figure) and the correlation of hourly sediment temperatures with ebullition fluxes calculated for single measurements (figure below). The difference in $r^2$ between both correlations confirms the statement that ebullition events occur erratically and that *"periods characterized by more pronounced ebullition seemed to roughly follow the sediment temperature-driven $CH_4$ production within the sediment as e.g. reported by Bastviken et al. 2004 (Fig. 5 (now 6))."*as stated within the MS.

[Figure]

33. p.10, line 308 Does the contribution of ebullition to the total flux (in %) also exhibit a diurnal pattern?
Yes. However, as shown in Fig. 5b (now 6b), ebullition itself did not show any clear systematic or diurnal trend. The contribution of ebullition (in %) to total (hourly) $CH_4$ emissions only shows diurnal patterns, because of diffusive $CH_4$ fluxes (evidencing diurnal patterns) being subtracted from the total CH4 flux. Hence, the diurnal trend in the contribution (in %) of ebullition to total CH4 fluxes is solely due to dynamics found for the diffusive flux component. Absolute ebullition fluxes do not show any diurnal trend.

34. Fig. 2 This Figure would benefit from annotations (e.g. the fan, the chamber, water tub). "Injections of gaseous mixture amounted to ..." - this information is not relevant in the caption and is already given in the text.

We agree and added annotations for the fan, the chamber, the water-filled tub, the sealed frame and vent (see below). The sentence *"Injections of gaseous mixture amounted to ..."* was removed from the figure caption.

[Figure]

35. Fig. 4 This Figure is not very readable and very complex. To make it easier for the reader to understand the Figure, I suggest the following changes: Data points should be bigger, it is almost impossible to distinguish open and black circles. Axis labels should be bigger. The Figure needs a legend that allows the reader to see what the dashed/solid lines and open/black circles denote without having to read the caption. At the same time, if this legend is included, you can remove the extensive and somewhat complicated descriptions of dashed/solid lines, open/black circles in the caption. Why was no death band applied in a and c?

We agree and enhanced the size of the data points and axis labels. In addition we included a legend explaining the dashed and solid line in Fig. 3a/3b and the filled dots and dashed lines in Fig. 3c/3d. We also shortened the figure caption. Furthermore, we included the death band and the data points removed due to the death band to a and c as well (please see figure below).

*" Time series plot of recorded concentrations (ppm) within the chamber headspace for (a) a simulated ebullition event and (b) an exemplary field study $CH_4$ measurement. Time spans dominated by diffusive $CH_4$ release are marked by (c-d) black dots, enclosed by the 25 % and 75 % quantiles $\pm$ 0.25 IQR of obtained concentration changes, shown as black dashed lines. Unfilled dots outside the dashed lines display ebullition events (see also Goodrich et al. 2011; Miller and*

*Oremland 1988). Gray shaded areas indicate the applied deathband at the beginning of each measurement (25%). Negative ΔCH₄ values indicate a overcompensation due to (temporally) insufficiant headspace mixing."*

[Figure]

36. Fig. 3 The data points should have error bars. The axis limit could be reduced to 7. If r2 is shown, I'd suggest to also show p and the calculated slope and intercept of the regression line.

We reduced the axis limits to 6 and added p-value, slope and intercept (please see figure below). Errors can be only calculated for the diffusive flux component, but not the total flux, which is calculated using the difference between the start and end $CH_4$ concentration. This prevents an error calculation for single ebullition events. As a result of this no error bars can be given for calculated ebullition events within this figure.

[Figure]

37. Fig. 5 This is a very interesting way to present your data. However, similar to Fig. 3, the Figure is very complex and not easily readable. I would like to suggest bigger labels, and a legend like I said in my comment above. A general question, does the bottom slice of the pie (i.e. 6 o'clock on a normal clock) correspond to 12 o'clock noon? If this is correct, then maybe it is good to warn the reader that what he normally perceives as 6 o'clock is not 6 o'clock in this Figure. I think it would be a good idea to have an "example clock-pie" with the actual hours (Let's say, 0:00, 6:00, 12:00, 18:00 ) next to the Figure so that it is easier to understand the clock-concept at first glance, otherwise the clock-concept might be a bit misleading.

We agree and enhanced the size of all labels and the axis titles. We decided to not include a legend, because the figure is already quiet complex. Yes, the bottom slice of the pie (i.e. 6 o'clock on a normal clock) corresponds to 12 o'clock noon. We really appreciate the remark and included a bigger slice at the top left of figure c, were the concept of the 24-hours-clock is shown now (please see figure below).

[Figure]

**Technical corrections**

38. p. 2, line 46 change to "given in the literature"
    Done.

39. p.3, line 77 What does "at all scales" refer to?
    We removed this part of the sentence. It was intended to make clear that ebullition events occur erratically during short (seconds to minutes) but also longer periods (hours to days or even more), as also stated by Anthony and Anthony (2013) who wrote that *"ebullition is episodic"* with *"frequencies of several minutes to weeks depending on the seep type, atmospheric pressure*

*dynamic, and season of year"* and that the size of bubbles emitted to the atmosphere during a ebullition event is different.

40. p.5, line 148 change to "in the beginning"
    Done.

41. p.5, line 150 change to "were reported"
    Done.

42. p.5, line 156 "below the chambers"
    Done.

43. p.7, line 203 "outlier"
    Done.

44. p. 7, line 222 I am unsure about "reasonable controlled conditions". I'd suggest to delete "reasonable"?
    Done.

45. p. 8, line 232 change to "were calculated"
    Done.

46. p. 8, line 262 "explanatory approaches could be addressed" - I think the wording needs to be changed here.
    We removed this sentence from the MS due to made changes within section 3.

47. p. 9, line 286 "This dynamics are ..." should be changed to "These dynamics are"
    Done.

48. p. 9, line 291 daytime and nighttime are sometimes written as day time and night time throughout the text (here it is just most obvious because there are two different versions in the same sentence). Please check the article for consistent spelling of those terms.
    Changed to *"daytime"* and *"nighttime"* throughout the entire MS.

49. p.13, line 393 I think the title of that publication should be "Automated modeling of ecosystem $CO_2$ fluxes based on periodic closed chamber measurements: ..."
    We corrected the reference title.

**References (not in the MS):**

Anthony and Anthony, K. M. W., Anthony, P.: Constraining spatial variability of methane ebullition seeps in thermokarst lakes using point process models. Journal of Geophysical Research: Biogeosciences 118, 1015-1034, 2013.

Cohen , J.: A power primer. Psychological Bulletin 112(1), 155-159, 1992.

Field, A., Miles, J., Field, Z.: Discovering statistics using R. 992, 2000.

Fisher, R. A.: Statistical methods and statistical inference. 504p., 1956.

Franz, D., Koebsch, F., Larmanou, E., Augustin, j., Sachs, T.: High net $CO_2$ and $CH_4$ release at a euthropic shallow lake on a formerly drained fen. Biogeosciences13, 3051-3070, 2016.

Gazovic, M., Kutzbach, L., Schreiber, p., Wille, C., Wilmking, M.: Diurnal dynamics of CH from a boreal peatland during snowmelt. Tellus 62b, 133-139, 2010.

Hahn-Schöffl, M., Zak, D., Minke, M., Gelbrecht, J., Augustin, J., Freibauer, A.: Organic sediment formed during inundation of a degraded fen grassland emits large fluxes of $CH_4$ and $CO_2$. Biogeosciences 8, 1539-1550, 2011.

Hoffmann, M., Jurisch, N., Garcia Alba, J., Albiac Borraz, E., Schmidt, M., Huth, V., Rogasik, H., Rieckh, H., Verch, G., Sommer, M., Augustin, J.: Detecting small-scale spatial heterogeneity and temporal dynamics of soil organic carbon (SOC) stocks: a comparison between automatic chamber-derived C budgets and repeated soil inventories. Biogeosciences Discussion, doi: 10.5194/bg-2016-332, 2016.

Lindgren, P. R., Grosse, G., Anthony, K. M. W., Meyer, F. J.: Detection and spatiotempral analysis of methane ebullition on thermokarst lake ice using high-resolution optical aerial imagery. Biogeosciences 13, 27-44, 2016.

Pohl, M., Hoffmann, m., Hagemann, U., Giebels, m., Albiac Borraz, E., Sommer, M., Augustin, J.: Dynamic C and N stocks-key factors controlling the C gas exchange of maize in a heterogenenous peatland. Biogeosciences 11, 2737-2752.

Steffenhagen, P., Zak, D., Schulz, K., Timmermann, T., Zerbe, S.: Biomass and nutrient stock of submerged and floating macrophytes in shallow lakes formed during rewetting of degraded fens. Hydrobiologia 692, 99-109, 2012.

---

## Author Comment (AC2) · 28 Oct 2016

**Answers to Anonymous Referee #2**

**General comments**

1. This paper […] would be a useful contribution to enhance our current understanding of CH4 dynamics in wetland systems, but the paper needs to be further revised before being considered for publication in this journal. As this paper aims to describe a method in determining ebullition and diffusion from the concentration trace of autochamber measurements, the authors should spend more time in describing the methodological details of the calculation (e.g. determination of diffusive flux) and justifying the use of this approach compared to other existing ones (e.g. Goodrich et al 2011) in estimating ebullition.

   We added a flow chart showing the presented algorithm and a more detailed explanation of it to section *"2.2 Flux calculation and separation algorithm"*. We also added a more detailed discussion about advantages compared to other direct or indirect separation approaches (justification) to section *"3.3 Overall performance"* (please see also 24/28. Comment reviewer #1).

2. The authors spent a substantial proportion of time examining the temporal variability of diffusive, ebullition, and total CH4 fluxes – but these does not prove that the algorithm is working successfully. There is a need to provide further validation of this method in separating diffusion and ebullition through field testing, e.g. the use of bubble traps. Such comparison should be done in this paper to provide a more affirmative testing of the algorithm, rather than in future studies as suggested by the authors in the last paragraph.

   The laboratory experiment showed than the algorithm works to filter ebullition events. In addition we now refer to eddy covariance measurements at the same field site during the same study period, showing comparable total $CH_4$ emissions (please see also answer to 23. Comment reviewer #1). We think that Bubble traps are not providing a sufficient validation of short term or single ebullition events occurring at the measurement. On the one hand, measurements of total (automatic chamber) and ebullition flux (bubble traps) would be spatially separated. On the other hand, a number of authors showed that ebullition events occur erratically in time and space (e.g. Lindgren et al. 2016; Anthony and Anthony 2013; see pictures below showing bubbles trapped in ice at the study site during January 2016). Thus, the spatial and temporal difference between separated AC measurements and bubble trap measurements will introduce a spatial and temporal error into the comparison of calculated fluxes using the presented approach and measurements of e.g. bubble traps. This bias can be only reduced by a sufficient number of spatial (bubble traps as well as automatic chambers) and temporal measurement repetitions and data aggregation. Both requirements, needed for a low bias will, however, only yield in a comparison of either spatially or temporally (month to years) aggregated comparison, but are not a proper validation of single ebullition events or short term separations of ebullition and diffusion (e.g. hours or days). If the bubble traps, however, would be installed underneath the chambers, to overcome mentioned spatial and temporal heterogeneity, the chamber measurements would be substantially biased, since they would not include ebullition events trapped by the bubble traps. We therefore do not agree on the statement that bubble trap measurements are a suitable method for validating the accuracy of the presented calculation algorithm regarding single measurement or short term (<1 year) flux separation results.

[Figure]

[Figure]

To better address this issue and show the huge advantage of the presented approach we added a discussion regarding direct (e.g. bubble traps) and indirect (data processing based approaches) methods to estimate the CH$_4$ flux components to section *"3.3 Overall performance"* (please see also answers to 24/28. comment reviewer #1*).

3. Also, are there any drawbacks of using IQR of concentration change to detect ebullition events? If ebullition occurs continuously through the measurement period, would the proposed method fail to identify ebullition events due to a consistently, large magnitude of concentration change?

As stated in the MS: *"... flux separation might be hampered due to a steady flux originating from other processes than diffusion through peat and water layers, such as the steady ebullition of micro bubbles (Prairie and del Giorgio 2013; Goodrich et al. 2011)."*. A consistently large magnitude in concentration changes during the chamber measurements will not hamper the total flux calculation. However, flux separation might be hampered. Out of 14828 valid measurements the algorithm was unable to calculate a valid diffusive flux for 170 measurements. This equals less than 1.15 % of all measurements. Taken into account that the presented measurement site is characterized by rather large CH4 emissions (e.g. Franz et al. 2015) and consistently occurring ebullition events, this problem seems to be negligible for the presented site, but might be of relevance in other wetland ecosystems. We therefore stated the need to apply presented algorithm to other wetland ecosystems within section *"4. Conclusions"* of the MS.

4. Also would the proposed method be able to identify both "major" and "minor" ebullition events? The lab test does not provide a definitive answer to these questions, and further tests with a greater variety of conditions (e.g. pulse vs. continuous injection) are required.

Due to the variable ebullition filter (IQR-filter), the proposed method identifies is able to identify "minor" and "major" ebullition events. This is also shown by the laboratory experiment which includes simulated ebullition events of different strength. However, ebullition flux components are given as flux integrated over the entire measurement (which might include bigger and smaller ebullition events not separated from each other). Calculated ebullition fluxes (integrated over a single measurement) ranged from approx. zero (0.0002 µmol m$^{-2}$ s$^{-1}$) to 0.6780 µmol m$^{-2}$ s$^{-1}$, which supports the capability of the algorithm to identify "minor" and "major" ebullition events.

5. Since the algorithm only estimates ebullition as the difference between total CH4 flux and diffusive flux, accurate quantification of the diffusive components becomes very important. What is the minimum detectable flux of this system?

The precision of the sensor for CH$_4$ measurements is 0.6 ppb (10sec records). In principle the minimum detectable flux would be a (near) zero flux. The lowest valid diffusive flux measured during the study period was 0.0012 µmol m$^{-2}$ s$^{-1}$, which equals a concentration change over 10 minutes of 1.14 ppb (this detection limit might change due to the user defined parameter setups made in the script (e.g. set p-value)).

6. In open-water systems, CH4 flux is expected to be lower than that in vegetated wetlands – if CH4 concentration change within the 15-second interval is not observable, the calculation of diffusive flux might be biased that further causes inaccuracies in the estimation of ebullition. Why not determine the ebullition component directly from the time series of headspace CH4 concentration? These issues should be addressed before the algorithm could be trusted and applied in other autochamber systems for separating the CH4 flux components.

In the case of low concentration changes the calculated diffusive $CH_4$-flux using the presented approach would be around zero. The reason therefore is that the calculation algorithm is not based on the $r^2$ as quality criteria, which was e.g. used by Goodrich et al. (2011), who discarded fluxes for steady flux analysis (diffusion) with an $r^2 < 0.8$. A flux will be identified as long as the regression slope is significant. In the case of low but valid diffusive fluxes, the potential bias to separated fluxes will be small as well and the influence on the ebullition flux component rather negligible. If diffusive flux calculation is not possible due to invalid fluxes (non-significant regression slope), the measurement system (e.g. chamber design or measurement frequency and duration) could be adapted (e.g. smaller chamber volume with bigger basal area), since non-observable fluxes are rather a problem of the used measurement device (chamber system) than the presented flux calculation and separation approach (which can be only as good as the data it is based on). Thus the detection limit might be reduced.

**Specific comments**

7. L41-42 – Were the quartiles and IQR referring to concentrations within one measurement period?

Yes. The IQR is referring to the entire single measurement (without death band) and the quartiles are referring to the specific MW subset of the single measurement. If the IQR would refer to the specific MW, the diffusive $CH_4$ emission present within subsections of the measurement without ebullition events could be not identified due to the highly sensitive IQR-criteria.

If the quartiles would refer to the entire MW, a bigger ebullition event during the measurement would result in a diffusive flux component, which includes smaller ebullition events, cause the filter criteria would be not sensitive enough.

8. L70 – In open-water systems, I assume vegetation is absent. If this is the case, plant-mediated transport will not be one of the CH4 release mechanisms.

We agree and changed the sentence to:

*"In wetland ecosystems, $CH_4$ is released via three main pathways: I) diffusion (including "storage flux", in terms of rapid diffusive release from methane stored in the water column), ii) ebullition and iii) plant-mediated transport (e.g., Goodrich et al. 2011; Bastviken et al. 2004; Van der Nat and Middelburg 2000; Whiting and Chanton 1996)."*

9. L108 – How does one define "medium" and "major" ebullition events? Are there any objective criteria for such categorization?

We changed the sentence by adding the threshold given by Goodrich et al. (2011): *"However, the static threshold to determine ebullition events, as well as low-resolution measurements, limited the approach to estimates ebullition events characterized by a sudden concentration increase $\geq 8$ nmol $mol^{-1}$ $s^{-1}$, which prevents a clear flux separation."*. The threshold given in Goodrich et al. equals an increase due to ebullition of 1.273 ppm per 15s record for the measurement system presented in our MS (bigger chamber volume). Thus a number of smaller ebullition events would be not taken into account. However, it is of course hard to give objective criteria for our categorization of *"major"*, *"medium"* or *"minor"* ebullition events. These categories might be different from measurement site to measurement site as well as the used chamber design (size and

volume). This is the main reason why we want to introduce a variable threshold for flux separation.

10. L134-136 – I think the hypothesis can be further refined. The flux separation algorithm might help tease out the contribution of diffusion and ebullition to overall flux, but itself could not be used to reveal the spatial and temporal dynamics – this is rather achieved by the AC system.
We partly agree. Both, the chamber system and the used algorithm complement each other (please see also answer to 26. Comment reviewer #1 and changes in 3.3). Spatial dynamics are of course a result of the spatially distributed chambers and not of the flux separation algorithm. However, temporal dynamics for the diffusive and ebullition flux components are obtained through the presented algorithm, since the chamber measurements itself only deliver temporal dynamics of total $CH_4$ fluxes. Therefore we modified our hypothesis to:

*"We hypothesize that the presented flux calculation and separation algorithm can reveal together with the presented AC system, concealed spatial and temporal dynamics in ebullition- and diffusion-associated $CH_4$ fluxes. This will facilitate the identification of relevant environmental drivers.".*

11. L161-165 – Only 4 chambers along the transect? No replications? Also, what is the shape of the chamber?
Yes, the study site consist of 4 Chambers along a transect. However, the spatial difference between the chambers along the transect is not within the scope of this MS, since the study site only acts as test data set, wherefore 4 chambers are assumed to be sufficient. The same accounts for replications. The chamber shape is shown in Fig. 1 and now also stated in the MS (section 2.1):

*"The AC system consists of four rectangular transparent chambers, installed along a transect from the shoreline into the lake."*

12. L171-173 – What is the rate of gas flow within the AC system? Do the chambers equipped with a vent tube for pressure equalization?
We added the gas flow rate of 5 l per minute to the MS. The chambers are not equipped with a vent for pressure equalization. Possible effect due to this measurement method specific limitation were tried to be reduced by a rather slow and soft chamber closure as well as by the applied death band at the beginning of each measurement. This might be a limitation of the presented AC system (chamber system), but it is not a limitation for the presented flux separation algorithm. Quiet the converse, the good overall agreement of the in parallel performed eddy covariance measurements (please see also answer to 23. comment reviewer #1) with the shown chamber measurements of the total CH4 emissions indicate that either the chamber measurements are not affected, or that the flux calculation approach was able to discard pressure related artifacts.

13. L173-175 – What did "overcompensation" exactly mean? Did you refer to the drop in CH4 concentration in the chamber headspace – this looked strange to me.
Yes exactly. We added a short explanation for this to the MS (please see also answer to 1. comment reviewer #1).
Would this mean that the fan is not effective enough in homogenizing the headspace air?
Yes, this means that the fan is homogenizing the comparable huge chamber volume (1.5 $m^3$) within 15 to 30 sec.  A stronger fan might trigger ebullition by stir up the water underneath the chamber. However, as stated in the MS: *"To avoid measurement artefacts (e.g., overcompensation), being taken into account as start or end concentration, measurement points representing an inherent concentration change smaller or larger than the upper and lower*

*quartile ± 0.25 times IQR were discarded prior to calculation of the total CH₄ flux."*, the presented algorithm is accounting for this artifact. Since the chamber headspace homogenization is a purely chamber design related limitation, it is not a limitation of the presented flux separation algorithm. A faster homogenization would simply result in a non-discarding of concentration records for the total flux calculation, and thus not affecting the flux separation result.

14. L192-196 – The equation deriving CH4 flux does not look right – the unit of CH4 flux shown is umol C m-2 s-1, but in the calculation molar mass of CH4 is used?
We corrected *"μmol"* into *"μg"* within the equation.

15. L200-204 – Would appreciate a more in-depth description of the protocol here, as this is the crucial part of the paper. How did the variable moving window work? Were fluxes calculated for various durations of MW within the 10-min deployment period, as long as the rigid outlier test was passed? If this was the case, which one would be chosen to represent diffusive fluxes?
Yes, various diffusive fluxes might be calculated for one flux measurement using different durations of this measurement. We added a flow chart to the MS, to better explain the used algorithm (please see also answer to 1. comment reviewer #1). Furthermore, we rewrote section 2.2 and added more details.

*"The resulting numerous $CH_{4_{diffusion}}$ fluxes calculated per measurement (based on the moving window data subsets) were further evaluated according to different exclusion criteria: (i) range of within-chamber air temperature not larger than ± 1.5 K; (ii) significant regression slope (p ≤ 0.1, t-test); and (iii) non-significant tests (p > 0.1) for normality (Lillifor´s adaption of the Kolmogorov-Smirnov test), homoscedasticity (Breusch-Pagan test) and linearity. In addition (iv) abrupt concentration changes within each MW data subset were identified by a rigid outlier test, which discarded fluxes with an inherent concentration change outside of the range between the upper and lower quartile ± 0.25 times (user defined) the interquartile range (IQR). Calculated $CH_{4_{diffusion}}$ fluxes which did not meet all exclusion criteria were discarded. In case of more than one flux per measurement meting all exclusion criteria, the $CH_{4_{diffusion}}$ flux with a starting $CH_4$ concentration being closest to the atmospheric $CH_4$ concentration was chosen."*

16. L226 – Were these the volume and area of the chamber or the tub? How much water was added into the tub?
We added 12 l water to the tub. The volume of 0.114 m³ represents the total headspace volume (air-filled) of the construction. Tub and chamber volume were each 0.063 m³. To better address this, we changed the sentence to:

*"In order to artificially simulate ebullition events, distinct amounts (5, 10, 20, 30 and 50 ml) of a gaseous mixture (25 000 ppm $CH_4$ in artificial air; Linde, Germany) were inserted by a syringe through a pipe into a water filled tub (12 l) covered with a closed chamber (headspace V=0.114 m³; A= 0.145 m²)."*

17. L231-233 – This assumed that all the added gases would be released as gas bubbles without any CH4 being dissolved in water. How would the authors ensure the absence of dissolved CH4 in water?
Right, some of the injected CH4 will be dissolved in the water. We therefore added the following sentence: *"To ensure CH4-saturation after the first simulations of ebullition events, the water within the tub was not replaced during the laboratory experiment.".* The water temperature during the lab experiment was approx. 20°C, which results in a solubility for CH4 of ~25mg/l or 300 mg for the 12 l of added water. We injected 4.5 to 51.5 ml of a gaseous mixture with 25 000 ppm. This equals an injection of 115 to 1290 ppm/l per simulated ebullition event (n=20). Since we did

not change the water during the entire measurement, it was assumed that the water was saturated with CH4 after the first simulations (starting with a 50 ml injection). However, of course the issue of dissolved CH4 might be a part of the (small) deviation of the measured ebullition fluxes from the theoretical 1:1-agreement shown in Fig. 3 (now 4).

18. L237-239 – How long was the chamber closed, and how was CH4 ebullition converted to amount (mg m-2) as shown in Fig. 4? Would be useful to show the time series of CH4 concentration as well in the lab test.
Since the laboratory experiment was carried out manually, the closure time for the performed chamber measurements varied between 5 to 10 minutes. An exemplary time series of measured $CH_4$ concentrations during one pulse experiment is shown in Fig. 2a (now 3a). The expected concentration changes within the chamber headspace as the result of methane injections were calculated as the mixing ratio between the amount of inserted gaseous mixture (25 000 ppm) and the air filled chamber volume (2 ppm) and related to the chamber/tub basal area.

19. L242-246 – However, increasing the frequency of concentration measurement might make it harder to detect significant concentration changes for quantifying diffusive fluxes, which could be low in open-water systems.
This important issue is clearly a question of measurement accuracy and detection limits. Thus, it is not directly a problem of the presented calculation and flux separation algorithm. The presented chamber system has a rather large chamber volume, which makes it hard to observe really small concentration changes within a shorter period. However, applying the algorithm to a chamber system, featuring a smaller chamber volume might solve this problem. Hence, low fluxes are a limitation of the measurement system and its specific design (which shows the importance of adapting the measurement system to the measurement site conditions), but does not constitute a limitation of the presented algorithm (which can be only as good as the data it is based on).

20. L251-257 – Would need some elaborations on why this method is better than other existing methods of quantifying ebullition (e.g. Goodrich et al. 2011). Gas traps should still be able to work in shallow water systems?
We added advantages compared to direct measurements to section 3.3. Gas traps will work as long as the water level is not dropping below a certain level (given by their specific design). The water level at the presented study site, however, dropped several times during the last 10 years below the sediment surface, or stayed only a few cm (<5 cm) above the sediment surface. Hence, during this period the use of bubble traps and especially bubble shields will be limited depending on their specific design. A water level just slightly above the sediment/peat, however, will not necessarily exclude ebullition events (e.g. GAZOVIC et al. 2010). As a result, measurement gaps will occur when using bubble traps instead of the presented algorithm (please see as well section 3.3 given above).

21. L257-260 – It is a bit far-fetching to suggest that this method is "applicable to a broader range of different manual and automatic closed chamber systems, instrumental setups, study designs, and ecosystems" without other solid evidence.
We changed the sentence to:

*"As a result of this, the presented, data processing based approach will be applicable as long as the underlying closed chamber measurements deliver continuous data sets for $CH_4$ concentration and air temperature.*
*Accounting for the few prerequisites (high resolution closed chamber measurements) as well as mentioned advantages, an application of the presented approach to open-water areas of a broad range of wetland ecosystems and closed chamber systems is stated. ".*

22. L280-281 – Not sure about the claim that diffusive flux shifted to a daytime maximum was valid. Higher CH4 flux was still observed during the night period between midnight and 6 am.
We changed *"daytime"* for *"early morning hours"* within the paragraph.

23. L281-286 – This might be tested by measuring CH4 concentration at different water depths. While thermal mixing might be weaker during daytime, this might be compensated by stronger wind and mechanical mixing. Low wind speed at night might contribute to lower diffusive fluxes owing to poor mixing of air above water surface. Also, temperature in July and September did not differ that much – why was the diurnal pattern different between these two months? Further discussion is needed.
Actually the diurnal cycle changed twice (as shown in Fig. 6 (now 7)): once from July to August, and back again from August to September. We therefore rewrote the misleading sentences*:*

*"However, compared to the diurnal variability of the total CH$_4$ fluxes, a pronounced shift of maximum CH$_4$ emissions from early morning to nighttime hours was revealed for the diffusive flux component during August 2013 (Fig. 5 and 6 (now 6 and 7)).While maximum diffusive fluxes during July were recorded during early morning hours (approx. 3:00 to 6:00), a shift to the nighttime was observed for August (max. from 21:00 to 0:00). During September maximum fluxes shifted back to the early morning, with maximum fluxes between 0:00 and 9:00 (Fig. 6 (now 7))."*

24. Table 1 – No details about these statistical tests were given in the methodology section. How were differences among chambers tested – which post hoc test was used?
We added details about the statistical test (balanced case; Tukey HSD test), used to identify significant difference between the chamber positions to Tab. 1. The other used statistical tests/measures are indicated within the table caption. The decision on whether or not a dependency of the flux components from the different environmental variables is considered to be significant was made according to the average coefficient of determination from regression analysis of all daily data subsets. This was needed due to present seasonality in the flux data. That means that even though a dependency is indicated as being significant, it might be not true for some daily data subsets, and vice versa. The same accounts for given average NSE values, which might be higher but also lower for the different daily data subsets.
This is, however, of minor importance regarding the accuracy of the presented flux calculation and separation algorithm, for which the relative difference between found dependencies (for daily data subsets) in flux components compared to the total flux is more important.

25. Please justify the choice of $p < 0.1$ in detecting statistical significance – the norm is to use $p < 0.01$.
We decided to use $p < 0.1$, because field study measurements are usually related to a higher uncertainty (as shown, CH4 emissions at the study site are characterized by a high spatiotemporal heterogeneity), compared to laboratory experiments, wherefore it might be advantageous to use a more sensitive testing to detect potential environmental drivers (Type I error vs. Type II error). In addition, we do not agree on the statement of a standard or norm of $p < 0.01$. Different p-value (e.g. 0.05) are given in the statistical literature, without the aim to state a dogmatic standard p-value to be used (e.g. Field et al. (2000); Fisher (1956); Coehen (1992)).
Moreover, we were rather interested in showing the relative difference and not making a statement about the (absolute) significance of our results. That means that we intended to show two things: 1.) that environmental variables seems to influence the separated flux components differently (no clear driver for erratically ebullition events but dependencies for diffusive flux components) and 2.) that trends or tendencies observed for the total CH4 flux become more clear for the diffusive CH4 flux, when separating data noise originating from the ebullition flux component.

**Technical corrections**

26. L80-81 – "if aiming to identify relevant environmental drivers of CH4 emissions" is grammatically incorrect. Modify as "if relevant environmental drivers of CH4 emissions are to be identified"
Done.

27. L139 – "Exemplary" field data? Not sure about the purpose of this heading.
We changed this heading into *"2.4 Exemplary study site"*. This now includes the site description of the study site used to gain the test data set as well as a description of the performed environmental measurements.

28. L161 – Change "installed as transect" to "installed along a transect"
Done.

29. L175 – Fig. 3 appears below Fig. 2?
We checked, refreshed and corrected all figure and table references made in the MS.

30. L209-210 – The phrase "smaller or larger than the upper and lower quartile 0.25 times IQR" is confusing – do you mean something like outside of the range between the upper and lower quartile 0.25 times IQR?
Yes. To avoid confusion on this important issue, we changed the sentence to:

*"Abrupt concentration changes within the MW were identified by means of a rigid outlier test, discarding fluxes with an inherent concentration change outside of the range between the upper and lower quartile ± 0.25 times the interquartile range (IQR)".*

31. Figure 3 – This was not exactly a scatterplot of concentrations – perhaps a time series plot would be more appropriate.
We agree and changed the figure caption to:

*"Time series plot of recorded concentrations (ppm) within the chamber headspace for (a) a simulated ebullition event and (b) an exemplary CH$_4$ measurement."*

32. Figure 5 – Why would the bars (I assume is CH4 flux) in the top graph have different colors? What do gray and black colors represent (the bars, not the pie chart)?
Fig. 5a shows the measured total CH$_4$-flux as a subdivided bar diagram. This means that the grey part of the bar shows the respective ebullition flux component and the black part the diffusion flux component. Thus, it is possible to show the flux components proportions throughout the entire study period. To make this clearer we rewrote the figure caption:

*"**Fig. 6:** Time series of (a) total CH$_4$ emissions with proprotions of ebullition (grey bar) and diffusion flux components (black bar) during the study period from July until September 2013. Figure 5b and 5c show the seperated flux components ( (b) ebullition and (c) diffusion) togetehr with the development of important environmental parameters, which are assumed to explain their specific dynamics ((a) water level, (b) RH and wind speed and (c) sediment (solid line) and water temperature (dashed line)). Pie charts represent the biweekly pooled diurnal cycle of measured CH$_4$ fluxes. Slices are applied clockwise, creating a 24-hour clock, with black and light grey slices indicating hours with CH$_4$ flux above and below the daily mean, respectively.".*

33. Figure 6 – Please show the error bars for the data points. Also, change "Juli" to "July", and "montly" to "monthly"

We refreshed the figure (see below) and changed *"Juli"* to *"July"* and *"montly"* to *"monthly"* within the figure and figure caption. We now included the standard deviation (SD) as a gray shaded area (indicating the average value ± 1SD) around the shown development of monthly averaged diffusive CH4 emissions (please see figure below).

[Figure]

We didn't include this before, because we thought it might be misleading. As shown for the four weeks of August (please see figures below) and also within Fig. 5 (now 6) of the MS, the magnitude of calculated diffusive $CH_4$ emissions differs between the months but also the weeks and days of each month. As a result, and despite of the clear diurnal development during each single day, the SD for monthly averages of hourly diffusive $CH_4$ emission is rather high. This is however, a result of the temporal development in diffusive $CH_4$ emission during the study period and can't be used to qualify significant differences between day and night. A high SD is shown in particular for August 2013 and mainly a result of the first week of August. This week shows substantially higher diffusive $CH_4$ emissions (while still evidencing a diurnal development within these high emissions!) compared to the second, third and fourth week of August.

[Figure]

**References (not in the MS):**

Anthony and Anthony, K. M. W., Anthony, P.: Constraining spatial variability of methane ebullition seeps in thermokarst lakes using point process models. Journal of Geophysical Research: Biogeosciences 118, 1015-1034, 2013.

Cohen , J.: A power primer. Psychological Bulletin 112(1), 155-159, 1992.

Field, A., Miles, J., Field, Z.: Discovering statistics using R. 992, 2000.

Fisher, R. A.: *Statistical* methods and *statistical* inference. 504p., 1956.

Franz, D., Koebsch, F., Larmanou, E., Augustin, j., Sachs, T.: High net CO2 and CH4 release at a euthropic shallow lake on a formerly drained fen. Biogeosciences13, 3051-3070, 2016.

Gazovic, M., Kutzbach, L., Schreiber, p., Wille, C., Wilmking, M.: Diurnal dynamics of CH from a boreal peatland during snowmelt. Tellus 62b, 133-139, 2010.

Hahn-Schöffl, M., Zak, D., Minke, M., Gelbrecht, J., Augustin, J., Freibauer, A.: Organic sediment formed during inundation of a degraded fen grassland emits large fluxes of $CH_4$ and $CO_2$. Biogeosciences 8, 1539-1550, 2011.

Hoffmann, M., Jurisch, N., Garcia Alba, J., Albiac Borraz, E., Schmidt, M., Huth, V., Rogasik, H., Rieckh, H., Verch, G., Sommer, M., Augustin, J.: Detecting small-scale spatial heterogeneity and temporal dynamics of soil organic carbon (SOC) stocks: a comparison between automatic chamber-derived C budgets and repeated soil inventories. Biogeosciences Discussion, doi: 10.5194/bg-2016-332, 2016.

Lindgren, P. R., Grosse, G., Anthony, K. M. W., Meyer, F. J.: Detection and spatiotempral analysis of methane ebullition on thermokarst lake ice using high-resolution optical aerial imagery. Biogeosciences 13, 27-44, 2016.

Pohl, M., Hoffmann, m., Hagemann, U., Giebels, m., Albiac Borraz, E., Sommer, M., Augustin, J.: Dynamic C and N stocks-key factors controlling the C gas exchange of maize in a heterogenenous peatland. Biogeosciences 11, 2737-2752.

Steffenhagen, P., Zak, D., Schulz, K., Timmermann, T., Zerbe, S.: Biomass and nutrient stock of submerged and floating macrophytes in shallow lakes formed during rewetting of degraded fens. Hydrobiologia 692, 99-109, 2012.